# Polynomially Over-Parameterized Convolutional Neural Networks Contain Structured Strong Winning Lottery Tickets

**Arthur da Cunha**
Université Côte d'Azur, Inria, CNRS, I3S
Aarhus University
Aarhus, Denmark
dac@cs.au.dk

**Francesco d'Amore**
Aalto University
Bocconi University
Espoo, Finland
francesco.damore@aalto.fi

**Emanuele Natale**
Université Côte d'Azur, Inria, CNRS, I3S
Sophia Antipolis, France
emanuele.natale@inria.fr

## Abstract

The Strong Lottery Ticket Hypothesis (SLTH) states that randomly-initialised neural networks likely contain subnetworks that perform well without any training. Although unstructured pruning has been extensively studied in this context, its structured counterpart, which can deliver significant computational and memory efficiency gains, has been largely unexplored. One of the main reasons for this gap is the limitations of the underlying mathematical tools used in formal analyses of the SLTH. In this paper, we overcome these limitations: we leverage recent advances in the multidimensional generalisation of the Random Subset-Sum Problem and obtain a variant that admits the stochastic dependencies that arise when addressing structured pruning in the SLTH. We apply this result to prove, for a wide class of random Convolutional Neural Networks, the existence of structured subnetworks that can approximate any sufficiently smaller network.

This result provides the first sub-exponential bound around the SLTH for structured pruning, opening up new avenues for further research on the hypothesis and contributing to the understanding of the role of over-parameterization in deep learning.

## 1 Introduction

Much of the success of deep learning techniques relies on extreme over-parameterization [Zhang et al., 2017, 2021, Novak et al., 2018, Brutzkus et al., 2018, Du et al., 2019, Kaplan et al., 2020]. While such excess of parameters has allowed neural networks to become the state of the art in many tasks, the associated computational cost limits both the progress of those techniques and their deployment in real-world applications. This limitation motivated the development of methods for reducing the number of parameters of neural networks; both in the past [Reed, 1993] and in the present [Blalock et al., 2020, Hoefler et al., 2021].

Although pruning methods have traditionally targeted reducing the size of networks for inference purposes, recent works have indicated that they can also be used to reduce parameter counts during training or even at initialisation without sacrificing model accuracy. In particular, Frankle and Carbin [2019] proposed the *Lottery Ticket Hypothesis (LTH)*, which conjectures that randomly-initialised

37th Conference on Neural Information Processing Systems (NeurIPS 2023).

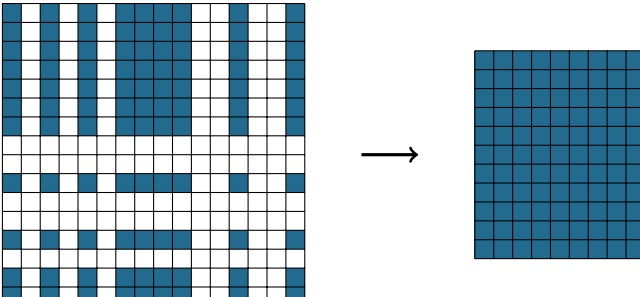

Figure 1: Illustration of neuron pruning. The left side shows the effect of pruning of neurons in the weight-matrix of a fully-connected layer. The rows in white correspond to neurons pruned in the associated layer while the columns in white represent the effect of removing neurons from the previous layers. On the right, we allude to the possibility of collapsing the pruned matrix into a smaller, dense one.

networks contain sparse subnetworks that can be trained and reach the performance of the fully-trained original network. Empirical investigations on the LTH [Zhou et al., 2019, Ramanujan et al., 2020, Wang et al., 2020] pointed towards an even more striking phenomenon: the existence of subnetworks that perform well without any training. This conjecture was named the *Strong Lottery Ticket Hypothesis (SLTH)* by Pensia et al. [2020].

While the SLTH has been proved for many different classes of neural networks (see Section 2), those works are restricted to unstructured pruning, where the subnetworks are obtained by freely removing individual parameters from the original network. However, this lack of structure can significantly reduce the gains that sparsity can bring, both in terms of memory and computational efficiency. The possibility of removing parameters at arbitrary points of the network implies the need to store the indices of the remaining non-zero parameters, which can become a significant overhead with its own research challenges [Pooch and Nieder, 1973]. Moreover, the theoretical computational gains of unstructured sparsity can also be difficult to realise in standard hardware, which is optimised for dense operations. Most notably, the irregularity of the memory access patterns can lead to both data and instruction cache misses, significantly reducing the performance of the pruned network.

The limitations of parameter-level pruning have motivated extensive research on *structured pruning*, which constrain the sparsity patterns to reduce the complexity of parameter indexation and, more generally, to make the processing of the pruned network more efficient. A simple example of structured pruning is *neuron pruning* of fully-connected layers: deletions in the weight matrix are constrained to the level of whole rows/columns. As illustrated by Figure 1, pruning under this constraint produces a smaller network that is still dense, directly reducing the computational costs without any need for extra memory to store indices. Similarly, deleting entire filters in Convolutional Neural Networks (CNNs) [Polyak and Wolf, 2015] or "heads" in attention-based architectures [Michel et al., 2019] also produces direct reductions in computational costs.

It is important to note that structured pruning is a restriction of unstructured pruning so, theoretically, the former is bound to perform at most as well as the latter. For example, by deleting whole neurons one can remove about 70% of the weights in dense networks without significantly affecting accuracy, while through unstructured pruning one can usually reach 95% sparsity without accuracy loss [Alvarez and Salzmann, 2016, Liu et al., 2019]. In practice, however, the computational advantage of structured pruning can offset this difference: a suitably structured sparse network can be more efficient than an even sparser one that lacks structure. This trade-off between sparsity and actual efficiency has motivated the study of less coarse sparsity patterns since weaker structural constraints such as strided sparsity [Anwar et al., 2017] (Figure 2b) or block sparsity [Siswanto, 2021] (Figure 2c) are already sufficient to deliver the bulk of the computational gains that structured can offer.

Despite its benefits, structured pruning has received little attention in the context of the SLTH. In fact, the work that first proved a version of the SLTH, Malach et al. [2020], has remained the only one to study this scenario, to the best of our knowledge. Moreover, it brings a negative result: the authors prove that removing neurons from a randomly-initialised shallow neural network (a single hidden layer) is equivalent to the random features model[1](e.g., Rahimi and Recht [2007, 2008]),

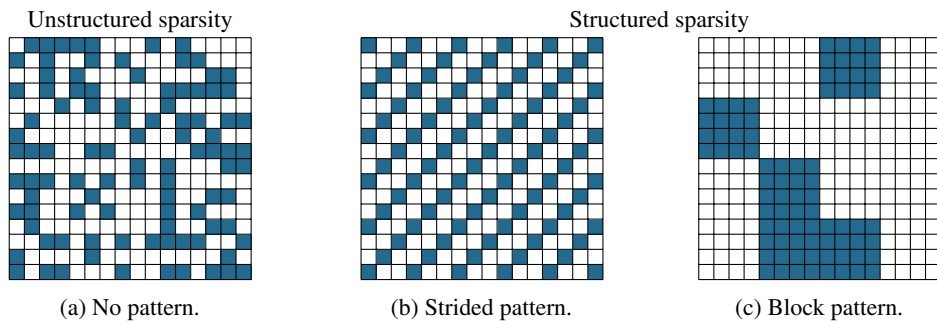

| Unstructured sparsity | Structured sparsity | |
| (a) No pattern. | (b) Strided pattern. | (c) Block pattern. |

Figure 2: Examples of different pruning patterns.

which cannot efficiently approximate even a single ReLU neuron [Yehudai and Shamir, 2019]. As most proofs around the SLTH involve pruning a shallow random network to build the desired approximations, those results show that the SLTH can be challenging to tackle when restricted to neuron pruning.

In addition, we believe that a factor hindering progress toward structured versions of the SLTH more generally is a limitation of a result underlying almost all of the theoretical works on the SLTH: a theorem by Lueker on the *Random Subset-Sum Problem (RSSP)*.

**Theorem 1** ([Lueker, 1998, Da Cunha et al., 2023]). *Let $X_1, \ldots, X_n$ be independent uniform random variables over $[-1, 1]$, and let $\varepsilon \in (0, 1/3)$. There exists a universal constant $C > 0$ such that, if $n \geq C \log(1/\varepsilon)$, then, with probability at least $1 - \varepsilon$, for all $z \in [-1, 1]$ there exists $S_z \subseteq [n]$ for which*

$$\left| z - \sum_{i \in S_z} X_i \right| \leq \varepsilon.$$

In general terms, the theorem states that given a rather small number of random variables, there is a high probability that any target value within an interval of interest can be approximated by a sum of a subset of the random variables. An important remark is that even though Theorem 1 is stated in terms of uniform random variables, it is not hard to extend it to a wide class of distributions.[2]

While Theorem 1 closely matches the setup of the SLTH, it only concerns individual random variables and directly applying it to entire random structures, as needed when considering structured pruning, would require an exponential number of random variables. The recent works by Borst et al. [2022], Becchetti et al. [2022] reduced this gap by proving multidimensional versions of Theorem 1. Still, the intricate manipulation of the network parameters in proofs around the SLTH imposes restrictions that are not covered by those results.

**Contributions**

In this work, we overcome those obstacles and prove that random networks in a wide class of CNNs are likely to contain structured subnetworks that approximate any sufficiently smaller CNN. To the best of our knowledge, our results provide the first sub-exponential bounds around the SLTH for structured pruning of deep neural networks of any kind. More precisely,

- We prove a multidimensional version Theorem 1 that is robust to some dependencies between coordinates, which is crucial for structured pruning (Theorem 5);

- We leverage this result and, by combining two types of structured sparsity (block and neuron/filter sparsity), we show that, with high probability and for a wide class of architectures, polynomially over-parameterized random networks can be pruned in a structured manner to approximate any target network (Theorem 2);

---

[1]In the random features model, one can only train the last layer of the network.

[2]Distributions whose probability density function $f$ satisfies $f(x) \geq b$ for all $x \in [-a, a]$, for some constants $a, b > 0$ (see Lueker [1998, Corollary 3.3]).

- Our results cover CNNs, which generalise fully-connected networks as well as many layer types commonly used in modern architectures, such as pooling and normalisation layers;
- Additionally, our pruning scheme focuses on filter pruning, which, like neuron pruning, allows for a direct reduction of the size and computational cost relative to the original CNN.

## 2   Related Work

**SLTH**   Put roughly, research on the SLTH revolves around the following question:

*Question.* Given an error margin $\varepsilon > 0$ and a target neural network $f_{\text{target}}$, how large must an architecture $f_{\text{random}}$ be to ensure that, with high probability on the sampling of its parameters, one can prune $f_{\text{random}}$ to obtain a subnetwork that approximates $f_{\text{target}}$ up to output error $\varepsilon$?

Malach et al. [2020] first proved that, for dense networks with ReLU activations, it was sufficient for $f_{\text{random}}$ to be twice as deep and polynomially wider than $f_{\text{target}}$. Orseau et al. [2020] showed that the width overhead could be greatly reduced by sampling parameters from a hyperbolic distribution. Pensia et al. [2020] improved the original result for a wide class of weight distribution, requiring only a logarithmic width overhead, which the authors proved to be asymptotically optimal. Da Cunha et al. [2022] generalised those results with optimal bounds to CNNs with non-negative inputs, which Burkholz [2022a] extended to general inputs and to residual architectures. Burkholz [2022a] also reduced the depth overhead to a single extra layer and provided results that include a whole class of activation functions. Burkholz [2022b] obtained similar improvements to dense architectures. Fischer and Burkholz [2021] modified many of the previous arguments to take into consideration networks with non-zero biases. Ferbach et al. [2022] further generalised previous results on CNNs to general equivariant networks. Diffenderfer and Kailkhura [2021] obtained similar SLTH results for binary dense neural networks within polynomial depth and width overhead, which Sreenivasan et al. [2022] improved to polylogarithmic overhead.

**Structured pruning**   Works on structured pruning date back to the early days of the field of neural network sparsification with works such as Mozer and Smolensky [1988] and Mozer and Smolensky [1989]. Since then, a vast literature has been built around the topic, particularly for the pruning of CNNs. For a survey of structured pruning in general, we refer the reader to the associated sections of Hoefler et al. [2021], and to He and Xiao [2023] for a survey on structured pruning of CNNs.

**RSSP**   Pensia et al. [2020] introduced the use of theoretical results on the RSSP in arguments around the SLTH, namely Lueker [1998, Corollary 3.3]. The work by Da Cunha et al. [2023] provides an alternative, simpler proof of this result. Borst et al. [2022] and Becchetti et al. [2022] proved multidimensional versions of the theorem. Theorem 5 diverges from those results in that it supports some dependencies between the entries of random vectors.

## 3   Preliminaries and contribution

Given $n \in \mathbb{N}$, we denote the set $\{1, \dots, n\}$ by $[n]$. The symbol $*$ represents the convolution operation, $\odot$ represents the element-wise (Hadamard) product, and $\phi$ represents the ReLU activation function. The notation $\|\cdot\|_1$ refers to the sum of the absolute values of each entry in a tensor. Similarly, $\|\cdot\|_2$ refers to the square root of the sum of the squares of each entry in a tensor. $\|\cdot\|_{\max}$ denotes the maximum norm: the maximum among the absolute value of each entry. Sometimes we represent a tensor $X \in \mathbb{R}^{d_1 \times \cdots \times d_n}$ by the notation $X = (X_{i_1, \dots, i_n})_{i_1 \in [d_1], \dots, i_n \in [d_n]}$. We denote the normal probability distribution with mean $\mu$ and variance $\sigma^2$ by $\mathcal{N}(\mu, \sigma^2)$. We write $U \sim \mathcal{N}^{d_1 \times \cdots \times d_n}$ to denote that $U$ is a random tensor of size $d_1 \times \cdots \times d_n$ with independent and identically distributed (i.i.d.) entries, each following $\mathcal{N}(0, 1)$. We refer to such random tensors as *normal tensors*. Finally, we refer to the axis of a 4-D tensor as *rows*, *columns*, *channels*, and *kernels* (a.k.a. filters), in this order.

For the sake of simplicity, we assume CNNs to be of the form $N \colon [-1, 1]^{D \times D \times c_0} \to \mathbb{R}^{D \times D \times c_\ell}$ given by
$$N(X) = K^\ell * \phi(K^{\ell-1} * \cdots \phi(K^1 * X)),$$
where $K^i \in \mathbb{R}^{d_i \times d_i \times c_{i-1} \times c_i}$ for $i \in [\ell]$, and the convolutions have no bias and are suitably padded with zeros. Moreover, when the kernels $K^{(i)}$ are normal tensors, we say that $N$ is a *random CNN*.

Our main result is the following.

**Theorem 2** (Structured SLTH). *Let $D, c_0, \ell \in \mathbb{N}$, and $\varepsilon \in \mathbb{R}_{>0}$. For $i \in [\ell]$, let $d_i, c_i, n_i \in \mathbb{N}$. Let $\mathcal{F}$ be the class of functions from $[-1, 1]^{D \times D \times c_0}$ to $\mathbb{R}^{D \times D \times c_\ell}$ such that, for each $f \in \mathcal{F}$*

$$f(X) = K^{(\ell)} * \phi(K^{(\ell-1)} * \cdots \phi(K^{(1)} * X)), \tag{1}$$

*where, for $i \in [\ell]$, $K^{(i)} \in \mathbb{R}^{d_i \times d_i \times c_{i-1} \times c_i}$ and $\left\| K^{(i)} \right\|_1 \leq 1$.*

*Let also $N_0 \colon [-1, 1]^{D \times D \times c_0} \to \mathbb{R}^{D \times D \times c_\ell}$ be a $2\ell$-layered random CNN given by*

$$N_0(X) = L^{(2\ell)} * \phi(L^{(2\ell-1)} * \cdots \phi(L^{(1)} * X)), \tag{2}$$

*where for $i \in [\ell]$ the kernels $L^{(2i-1)}$ and $L^{(2i)}$ are normal tensors of shape $1 \times 1 \times c_{i-1} \times 2n_i c_{i-1}$ and $d_i \times d_i \times 2n_i c_{i-1} \times c_i$, respectively.*

*Finally, let $\mathcal{G}$ be the class of subnetworks that can be obtained by pruning contiguous blocks of parameters and removing entire filters from $N_0$.*

*There exists a universal constant $C > 0$, such that if, for $i \in [\ell]$,*

$$n_i \geq C d_i^{13} c_i^6 \log^3 \frac{d_i^2 c_i c_{i-1} \ell}{\varepsilon},$$

*then, with probability at least $1 - \varepsilon$, we have that, for all $f \in \mathcal{F}$,*

$$\sup_{X \in [-1,1]^{D \times D \times c_0}} \min_{g \in \mathcal{G}} \|f(X) - g(X)\|_{\max} \leq \varepsilon.$$

The filter removals ensured by Theorem 2 take place at layers $1, 3, \ldots, 2\ell - 1$ and imply the removal of the corresponding channels in the next layer. The overall modification yields a CNN with kernels $\tilde{L}^{(1)}, \ldots, \tilde{L}^{(2\ell)}$ such that, for $i \in [\ell]$, the kernels $\tilde{L}^{(2i-1)}$ and $\tilde{L}^{(2i)}$ have shape $1 \times 1 \times c_{i-1} \times 2c_{i-1} m_i$ and $d_i \times d_i \times 2c_{i-1} m_i \times c_i$, respectively, where $m_i = \sqrt{n_i/(C_1 d_i \log(1/\varepsilon))}$ for a universal constant $C_1$. Moreover, our proof ensures that the kernels $\tilde{L}^{(2i-1)}$ can be required to have a specific type of block sparsity: they can be structured as if pruned by $2m_i$-channel-blocked masks, defined as follows.

**Definition 3** ($n$-channel-blocked mask). Given a positive integer $n$, a binary tensor $S \in \{0, 1\}^{d \times d \times c \times cn}$ is called $n$-channel-blocked if and only if

$$S_{i,j,k,l} = \begin{cases} 1 & \text{if } \left\lceil \frac{l}{n} \right\rceil = k, \\ 0 & \text{otherwise,} \end{cases}$$

for all $i, j \in [d]$, $k \in [c]$, and $l \in [cn]$.

We remark that, from a broader perspective, the central aspect of Theorem 2 is that the lower bound on the size of the random CNN depends only on the kernel sizes of the CNNs being approximated.

In subsection 4.2 we discuss the proof of Theorem 2. It requires handling subset-sum problems on multiple random variables at once (random vectors). Furthermore, the inherent parameter-sharing of CNNs creates a specific type of stochastic dependency between coordinates of the random vectors, which we capture with the following definition.

**Definition 4** (NSN vector). A $d$-dimensional random vector $Y$ follows a *normally-scaled normal* (NSN) distribution if, for each $i \in [d]$, $Y_i = Z \cdot Z_i$ where $Z, Z_1, \ldots, Z_d$ are i.i.d. random variables following a standard normal distribution.

A key technical contribution of ours is a Multidimensional Random Subset Sum (MRSS) result that supports NSN vectors. In subsection 4.1 we discuss the proof of the next theorem, which follows a strategy similar to that of [Borst et al., 2022, Lemmas 1, 15].

**Theorem 5** (Normally-scaled MRSS). *Let $0 < \varepsilon \leq 1/4$, and let $d$, $k$, and $n$ be positive integers such that $n \geq k^2$ and $k \geq C d^3 \log \frac{d}{\varepsilon}$ for some universal constant $C \in \mathbb{R}_{>0}$. Furthermore, let $X_1, \ldots, X_n$ be $d$-dimensional i.i.d. NSN random vectors. For any $\vec{z} \in \mathbb{R}^d$ with $\|\vec{z}\|_1 \leq \sqrt{k}$, there exists with constant probability a subset $S \subseteq [n]$ of size $k$ such that $\left\| \left( \sum_{i \in S} X_i \right) - \vec{z} \right\|_{\max} \leq \varepsilon$.*

While it is possible to naïvely apply Theorem 1 to obtain a version of Theorem 2, doing so leads to an exponential lower bound on the required number of random vectors.

# 4 Analysis

In this section, after proving our MRSS result (Theorem 5), we discuss how to use it to obtain our main result on structured pruning (Theorem 2). Full proofs are deferred to Appendix B.

## 4.1 Multidimensional Random Subset Sum for normally-scaled normal vectors

**Notation.** Given a set $S$ and a positive integer $n$, we denote by $\binom{S}{n}$ the family of all subsets of $S$ containing exactly $n$ elements of $S$. Given $\varepsilon \in \mathbb{R}_{>0}$, we define the interval $I_\varepsilon(z_i) = [z_i - \varepsilon, z_i + \varepsilon]$ and the multi-interval $I_\varepsilon(\vec{z}) = [\vec{z} - \varepsilon\mathbf{1}, \vec{z} + \varepsilon\mathbf{1}]$, where $\mathbf{1} = (1, 1, \ldots, 1) \in \mathbb{R}^d$. Moreover, for any event $\mathcal{E}$, we denote its complementary event by $\overline{\mathcal{E}}$.

In this subsection, we estimate the probability that a set of $n$ random vectors contains a subset that sums up to a value that is $\varepsilon$-close to a given target. The following definition formalises this notion.

**Definition 6** (Subset-sum number). Given (possibly random) vectors $X_1, \ldots, X_n$ and a vector $\vec{z}$, we define the $\varepsilon$-*subset-sum number* of $X_1, \ldots, X_n$ for $\vec{z}$ as

$$T_{X_1,\ldots,X_n}^k(\vec{z}) = \sum_{S \in \binom{[n]}{k}} \mathbf{1}_{\mathcal{E}_S^{(\vec{z})}},$$

where $\mathcal{E}_S^{(\vec{z})}$ denotes the event $\left\|\left(\sum_{i \in S} X_i\right) - \vec{z}\right\|_{\max} \leq \varepsilon$. We write simply $T_{n,k}$ when $X_1, \ldots, X_n$ and $\vec{z}$ are clear from the context.

To prove Theorem 5 we use the second moment method to provide a lower bound on the probability that the subset-sum number $T_{n,k}$ is strictly positive, which implies that at least one subset of the random vectors suitably approximates $\vec{z}$. Hence, we seek a lower bound on $\mathbb{E}[T_{n,k}]^2/\mathbb{E}[T_{n,k}^2]$.

Our first lemma provides a lower bound on the probability that a sum of NSN vectors is $\varepsilon$-close to a target vector, through which one can infer a lower bound on $\mathbb{E}[T_{n,k}]$.

**Lemma 7** (Sum of NSN vectors). *Let $k \in \mathbb{N}$, $\varepsilon \in \left(0, \frac{1}{4}\right)$, $\vec{z} \in \mathbb{R}^d$ such that $\|\vec{z}\|_1 \leq \sqrt{k}$ and $k \geq 16$. Furthermore, let $X_1, \ldots, X_k$ be $d$-dimensional i.i.d. NSN random vectors with $d \leq k$, and let $c_d = \min\left\{\frac{1}{d^2}, \frac{1}{16}\right\}$. It holds that*

$$\Pr\left(\sum_{i=1}^k X_i \in I_\varepsilon(\vec{z})\right) \geq \frac{1}{16}\left(\frac{2\varepsilon}{\sqrt{\pi\left(1 + 2\sqrt{c_d} + 2c_d\right)k}}\right)^d.$$

*Overview of the proof.* The main technical difficulty lies in the fact that the random vectors $X_1, \ldots, X_k$ are NSN vectors. Bounds can be easily derived for the case where the $X_i$ are i.i.d. normal random vectors by observing that the sum of normal random variables is also normal.

For $i \in [k]$, each entry of $X_i$ can be written as $Z_i \cdot Z_{i,j}$ where $Z_i$ and $Z_{i,j}$ are i.i.d. normal random variables. Conditional on $Z_1, \ldots, Z_k$, the $d$ entries of $X = \sum_{i=1}^k X_i$ are independent and distributed as $\mathcal{N}(0, \sum_{i=1}^k Z_i^2)$. By noticing that $Z_i^2$ is a chi-squared random variable and employing standard concentration inequalities (Lemma 17 in Appendix A) combined with the law of total probability, we can proceed as if the entries of $X$ were normal, up to some correction factors. $\square$

Bounding $\mathbb{E}[T_{n,k}^2]$ requires handling stochastic dependencies. Thus, we estimate the joint probability that two subsets of $k$ elements of $X_1, \ldots, X_n$ sum $\varepsilon$-close to the same target, taking into account that the intersection of the subsets might not be empty. The next lemma provides an upper bound on this joint probability that depends only on the size of the symmetric difference between the two subsets.

**Lemma 8** (Sum of NSN vectors). *Let $k, j \in \mathbb{N}_0$ with $1 \leq j \leq k$ Furthermore, let $X_1, \ldots, X_{k+j}$ be i.i.d. $d$-dimensional NSN random vectors with $k \geq Cd^3 \log\frac{d}{\varepsilon}$. Let $c_d = \min\left\{\frac{1}{d^2}, \frac{1}{16}\right\}$, $A = \sum_{i=1}^j X_i$, $B = \sum_{i=j+1}^k X_i$, and $C = \sum_{i=k+1}^{k+j} X_i$.[3] Then, it holds that*

$$\Pr\left(A + B \in I_\varepsilon(\vec{z}), B + C \in I_\varepsilon(\vec{z})\right) \leq 3\left(\frac{4\varepsilon^2}{\pi\left(1 - 2\sqrt{c_d}\right)j}\right)^d.$$

---

[3] We adopt the convention that $\sum_{i=1}^0 X_i = 0$.

*Overview of the proof.* Let $n = k + j$. We exploit once more the fact that, for all $i \in [n]$, each entry $X_i$ can be written as $Z_i \cdot Z_{i,j}$ where $Z_i$ and $Z_{i,j}$ are i.i.d. normal random variables. Conditional on $Z_1, \ldots, Z_{k+j}$, the $d$ entries of $A$, $B$, and $C$ are independent and distributed as $\mathcal{N}(0, \sum_{i=1}^{j} Z_i^2)$, $\mathcal{N}(0, \sum_{i=j+1}^{k} Z_i^2)$, and $\mathcal{N}(0, \sum_{i=k+1}^{k+j} Z_i^2)$, respectively. Hence, by the concentration inequalities for the sum of chi-squared random variables (Lemma 17 in Appendix A) and by the law of total probability, we can focus on the term

$$\Pr\left(A_i + B_i \in I_\varepsilon\left(z_i\right), B_i + C_i \in I_\varepsilon\left(z_i\right) \mid Z_1, \ldots, Z_n\right),$$

where $A_i$, $B_i$, and $C_i$ indicate the $i$-th entries of $A$, $B$, and $C$, respectively.

Another concentration argument for normal random variables (Lemma 14 in Appendix A), allow us to show that

$$
\begin{aligned}
\Pr\left(A_i + B_i \in I_\varepsilon\left(z_i\right), B_i + C_i \in I_\varepsilon\left(z_i\right) \mid Z_1, \ldots, Z_n\right) \\
&= \mathbb{E}_{B_i}\left[\Pr\left(A_i \in I_\varepsilon\left(z_i - B_i\right), C_i \in I_\varepsilon\left(z_i - B_i\right) \mid Z_1, \ldots, Z_n, B_i\right)\right] \\
&= \mathbb{E}_{B_i}\left[\Pr\left(A_i \in I_\varepsilon\left(z_i - B_i\right) \mid Z_1, \ldots, Z_n, B_i\right)\Pr\left(C_i \in I_\varepsilon\left(z_i - B_i\right) \mid Z_1, \ldots, Z_n, B_i\right)\right] \\
&\leq \mathbb{E}_{B_i}\left[\Pr\left(A_i \in I_\varepsilon\left(0\right) \mid Z_1, \ldots, Z_n, B_i\right)\Pr\left(C_i \in I_\varepsilon\left(0\right) \mid Z_1, \ldots, Z_n, B_i\right)\right] \\
&= \Pr\left(A_i \in I_\varepsilon\left(0\right) \mid Z_1, \ldots, Z_n\right)\Pr\left(C_i \in I_\varepsilon\left(0\right) \mid Z_1, \ldots, Z_n\right).
\end{aligned}
$$

Thus, we have reduced our argument to the estimation of probabilities of independent normal random variables being close to zero. □

The following lemma provides an explicit expression for the variance of the $\varepsilon$-subset-sum number.

**Lemma 9** (Second moment of $T_{n,k}$). *Let $k, n$ be positive integers. Let $S_0, S_1, \ldots, S_k$ be subsets of $[n]$ such that $|S_0 \cap S_j| = k - j$ for $j = 0, 1, \ldots, k$. Let $\mathrm{S}, \mathrm{S}'$ be two random variables yielding two subsets of $[n]$ of size $k$ drawn independently and uniformly at random. Let $X_1, \ldots, X_n$ be $d$-dimensional i.i.d. NSN random vectors. For any $\varepsilon > 0$ and $\vec{z} \in \mathbb{R}^d$, the second moment of the $\varepsilon$-subset sum number is*

$$\mathbb{E}\left[T_{n,k}^2\right] = \binom{n}{k}^2 \sum_{j=0}^{k} \Pr\left(|\mathrm{S} \cap \mathrm{S}'| = k - j\right) \Pr\left(\mathcal{E}_{S_0}^{(\vec{z})} \cap \mathcal{E}_{S_j}^{(\vec{z})}\right),$$

*where $\mathcal{E}_S^{(\vec{z})}$ denotes the event $\left\|\left(\sum_{i \in S} X_i\right) - \vec{z}\right\|_{\max} \leq \varepsilon$.*

*Proof.* Let $\mathrm{S}, \mathrm{S}'$ two random variables yielding two elements of $\binom{[n]}{k}$ drawn independently and uniformly at random. By the definition of $T_{n,k}$, we have that

$$
\begin{aligned}
\mathbb{E}\left[T_{n,k}^2\right] &= \mathbb{E}\left[\left(\sum_{S \in \binom{[n]}{k}} \mathbf{1}_{\mathcal{E}_S^{(\vec{z})}}\right)\left(\sum_{S' \in \binom{[n]}{k}} \mathbf{1}_{\mathcal{E}_{S'}^{(\vec{z})}}\right)\right] \\
&= \mathbb{E}\left[\sum_{S,S' \in \binom{[n]}{k}} \mathbf{1}_{\mathcal{E}_S^{(\vec{z})}} \mathbf{1}_{\mathcal{E}_{S'}^{(\vec{z})}}\right] \\
&= \sum_{S,S' \in \binom{[n]}{k}} \Pr\left(\mathcal{E}_S^{(\vec{z})} \cap \mathcal{E}_{S'}^{(\vec{z})}\right) \\
&= \sum_{S,S' \in \binom{[n]}{k}} \Pr\left(\mathcal{E}_\mathrm{S}^{(\vec{z})} \cap \mathcal{E}_{\mathrm{S}'}^{(\vec{z})} \mid \mathrm{S} = S, \mathrm{S}' = S'\right) \Pr\left(\mathrm{S} = S, \mathrm{S}' = S'\right) \\
&= \binom{n}{k}^2 \sum_{j=0}^{k} \Pr\left(\mathcal{E}_\mathrm{S}^{(\vec{z})} \cap \mathcal{E}_{\mathrm{S}'}^{(\vec{z})} \mid |\mathrm{S} \cap \mathrm{S}'| = k - j\right) \Pr\left(|\mathrm{S} \cap \mathrm{S}'| = k - j\right),
\end{aligned}
$$

as $\Pr\left(\mathcal{E}_\mathrm{S}^{(\vec{z})} \cap \mathcal{E}_{\mathrm{S}'}^{(\vec{z})}\right)$ depends only on the size of $\mathrm{S} \cap \mathrm{S}'$. □

**Overview of the proof of Theorem 5**

We use the second moment method (Lemma 15 in Appendix A) on the $\varepsilon$-subset-sum number $T_{n,k}$ of $X_1, \ldots, X_n$. Thus, we want to lower bound the right-hand side of

$$\Pr\left(T > 0\right) \geq \frac{\mathbb{E}[T_{n,k}]^2}{\mathbb{E}[T_{n,k}^2]}.$$

Equivalently, we can provide an upper bound on the inverse, $\frac{\mathbb{E}[T_{n,k}^2]}{\mathbb{E}[T_{n,k}]^2}$. By Lemma 9,

$$\mathbb{E}\left[T_{n,k}^2\right] = \binom{n}{k}^2 \sum_{j=0}^{k} \Pr\left(|\mathrm{S} \cap \mathrm{S}'| = k - j\right) \Pr\left(\mathcal{E}_{S_0}^{(\vec{z})} \cap \mathcal{E}_{S_j}^{(\vec{z})}\right)$$

where $\mathrm{S}, \mathrm{S}', S_i$ and $\mathcal{E}_S^{(\vec{z})}$ are defined as in the statement of the lemma. Observe also that

$$\mathbb{E}\left[T_{n,k}\right] = \sum_{S \in \binom{[n]}{k}} \mathbb{E}\left[\mathbf{1}_{\mathcal{E}_S^{(\vec{z})}}\right] = \sum_{S \in \binom{[n]}{k}} \Pr\left(\mathcal{E}_S^{(\vec{z})}\right) = \binom{n}{k} \Pr\left(\mathcal{E}_{S_0}^{(\vec{z})}\right).$$

By using the two above observations, we have

$$\frac{\mathbb{E}[T_{n,k}]^2}{\mathbb{E}[T_{n,k}^2]} = \frac{\binom{n}{k}^2}{\mathbb{E}\left[T_{n,k}\right]^2} \sum_{j=0}^{k} \Pr\left(|\mathrm{S} \cap \mathrm{S}'| = k - j\right) \Pr\left(\mathcal{E}_{S_0}^{(\vec{z})} \cap \mathcal{E}_{S_j}^{(\vec{z})}\right)$$

$$= \sum_{j=0}^{k} \Pr\left(|\mathrm{S} \cap \mathrm{S}'| = k - j\right) \frac{\Pr\left(\mathcal{E}_{S_0}^{(\vec{z})} \cap \mathcal{E}_{S_j}^{(\vec{z})}\right)}{\Pr\left(\mathcal{E}_{S_0}^{(\vec{z})}\right)^2}.$$

Lemma 7 provides a lower bound on the term $\Pr\left(\mathcal{E}_{S_0}^{(\vec{z})}\right)$ while Lemma 8 gives an upper bound on the term $\Pr\left(\mathcal{E}_{S_0}^{(\vec{z})} \cap \mathcal{E}_{S_j}^{(\vec{z})}\right)$.

In the full proof, we then show that $\Pr\left(|\mathrm{S} \cap \mathrm{S}'| \geq k/d\right)$ can be bounded using the Chernoff bound (Lemma 16 in Appendix A) even if we do not deal directly with binomial random variables. This allows us to discard the indices $j$ for which $\Pr\left(\mathcal{E}_{S_0}^{(\vec{z})} \cap \mathcal{E}_{S_j}^{(\vec{z})}\right)$ is large, which leads to the result after some technical manipulations.

## 4.2 Proving SLTH for structured pruning

To prove Theorem 2, we first show how to obtain the same approximation result for a single-layer CNN. Then, we iteratively apply the same argument for all layers of a larger CNN and show that the approximation error stays small.

We define the *positive* and *negative* parts of a tensor.

**Definition 10.** Given a tensor $X \in \mathbb{R}^{d_1 \times \cdots \times d_n}$, the *positive* and *negative* parts of $X$ are respectively defined as $X_{\vec{i}}^+ = X_{\vec{i}} \cdot \mathbf{1}_{X_{\vec{i}} > 0}$ and $X_{\vec{i}}^- = -X_{\vec{i}} \cdot \mathbf{1}_{X_{\vec{i}} < 0}$, where $\vec{i} \in [d_1] \times \cdots \times [d_n]$ points at a generic entry of $X$.

**Approximating a single-layer CNN**

We first present a preliminary lemma that shows how to prune a single-layer convolution $\phi\left(V * X\right)$ in a way that dispenses us from dealing with the ReLU $\phi$.

**Lemma 11.** *Let $D, d, c, n \in \mathbb{N}$ be positive integers, $V \in \mathbb{R}^{1 \times 1 \times c \times 2nc}$, and $X \in \mathbb{R}^{D \times D \times c}$. Let $S_1 \in \{0, 1\}^{\text{size}(V)}$ be a $2n$-channel-blocked mask. There exists a mask $S_2 \in \{0, 1\}^{\text{size}(V)}$ which only removes filters such that, for each $(i, j, k) \in [D] \times [D] \times [2nc]$, if $\tilde{S} = S_1 \odot S_2$, then*

$$\left(\phi\left(\left(V \odot \tilde{S}\right) * X\right)\right)_{i,j,k} = \left(\left(V \odot \tilde{S}\right)^+ * X^+ + \left(V \odot \tilde{S}\right)^- * X^-\right)_{i,j,k}.$$

*Overview of the proof.* $S_2 \in \{0,1\}^{\text{size}(V)}$ is such that $\tilde{V} = V \odot \tilde{S} = (V \odot S_1) \odot S_2$ contains only non-negative edges going from each input channel $t$ to the output channels $2(t-1)n+1, \ldots, (2t-1)n$, and only non-positive edges going from each input channel $t$ to the output channels $(2t-1)n+1, \ldots, 2tn$. Notice that, after applying the $2n$-channel-blocked mask $S_1$, the only possible non-zero entry of the $k$-th filter is located at its $t = \lceil \frac{k}{2n} \rceil$-th channel. Hence, for each $k \in [2nc]$, we keep filter $k$ if the entry of its $t$-th channel is non-negative and $2t - 1 = \lceil \frac{k}{n} \rceil$, or it is non-positive and $2t = \lceil \frac{k}{n} \rceil$. $\qquad\square$

We approximate a single convolution $K * X$ by pruning a polynomially larger neural network of the form $U * \phi(V * X)$ exploiting only a channel-blocked mask and filter removal: this is achieved using the MRSS result (Theorem 5).

**Lemma 12** (Kernel pruning)**.** *Let* $D, d, c_0, c_1, n \in \mathbb{N}$ *be positive integers,* $\varepsilon \in \left(0, \frac{1}{4}\right), M \in \mathbb{R}_{>0},$ *and* $C \in \mathbb{R}_{>0}$ *be a universal constant with*

$$n \geq C d^{13} c_1^6 \log^3 \frac{d^2 c_1 c_0}{\varepsilon}.$$

*Let* $U \sim \mathcal{N}^{d \times d \times 2nc_0 \times c_1}, V \sim \mathcal{N}^{1 \times 1 \times c_0 \times 2nc_0}$ *and* $S \in \{0,1\}^{\text{size}(V)},$ *with* $S$ *being a* $2n$-*channel-blocked mask. We define* $N_0(X) = U * \phi(V * X)$ *where* $X \in \mathbb{R}^{D \times D \times c_0},$ *and its pruned version* $N_0^{(S)}(X) = U * \phi((V \odot S) * X).$ *With probability* $1 - \varepsilon,$ *for all* $K \in \mathbb{R}^{d \times d \times c_0 \times c_1}$ *with* $\|K\|_1 \leq 1,$ *it is possible to remove filters from* $N_0^{(S)}$ *to obtain a CNN* $\tilde{N}_0^{(S)}$ *for which*

$$\sup_{X: \|X\|_{\max} \leq M} \left\| K * X - \tilde{N}_0^{(S)}(X) \right\|_{\max} < \varepsilon M.$$

*Overview of the proof.* Exploiting Lemma 11, for each $(r, s, t_1) \in [d] \times [d] \times [c_1]$, one can show that

$$\left( U * \phi\left( \left( V \odot \tilde{S}\right) * X\right)\right)_{r,s,t_1} = \sum_{i,j \in [d], t_0 \in [c_0]} \left( \sum_{k \in [nc_0]} U_{i,j,k,t_1} \cdot \tilde{V}^+_{1,1,t_0,k} \right) \cdot X^+_{r-i+1,s-j+1,t_0}$$

$$+ \sum_{i,j \in [d], t_0 \in [c_0]} \left( \sum_{k \in [nc_0]} U_{i,j,k,t_1} \cdot \tilde{V}^-_{1,1,t_0,k} \right) \cdot X^-_{r-i+1,s-j+1,t_0},$$

where $\tilde{S} = S_1 \odot S_2$, with $S_1$ being a $2n$-channel-blocked mask and $S_2$ being a mask that removes filters. Through a Chernoff bound, we show that $\tilde{V}^+_{1,1,t_0,:}$ has at least $n/3$ non-zero entries. Up to reshaping the tensor as a one-dimensional vector, we observe that $U_{:,:,k,:} \tilde{V}^+_{1,1,t_0,k}$ is an NSN vector (Lemma 18 in Appendix A). Hence, we can apply a boosted version of the MRSS result (Corollary 19 in Appendix A) and show that, with high probability, for all target filters $K$ with $\|K\|_1 \leq 1$, we can prune all but roughly $\sqrt{n/(C_1 d \log \frac{1}{\varepsilon})}$ positive entries of $\tilde{V}^+_{1,1,t_0,k}$, with $C_1$ being a universal constant, such that $\sum_{k \in [nc_0]} U_{:,:,k,:} \tilde{V}^+_{1,1,t_0,:}$ approximates the channels $K_{:,:,t_0,:}$ up to error $\varepsilon/(2d^2 c_0 c_1)$. The same holds for $\sum_{k \in [nc_0]} U_{:,:,k,:} \tilde{V}^-_{1,1,t_0,:}$. This pruning can be achieved by applying a third mask $S_3$ that only removes filters. Through some non-trivial calculations and by applying the Tensor Convolution Inequality (Lemma 20 in Appendix A), one can combine the above results to get the thesis. Notice that the overall pruning can be represented by the mask $S_1 \odot (S_2 \odot S_3)$, where $S_2 \odot S_3$ is a mask that only removes filters, and $S = S_1$ is a $2n$-channel-blocked mask. $\qquad\square$

*Remark* 13. From the proofs of Lemmas 12 and 11, we can see that the overall modification yields a pruned CNN $\hat{U} * \phi(\hat{V} * X)$ with $\hat{V} \in \mathbb{R}^{1 \times 1 \times c_0 \times 2mc_0}$ and $\hat{U} \in \mathbb{R}^{d \times d \times 2mc_0 \times c_1}$, where $m = \sqrt{n/(C_1 d \log \frac{1}{\varepsilon})}$ for a universal constant $C_1$. Moreover, the kernel $\hat{V}$ is obtained through a 3-stage pruning process: First, we apply a $2n$-channel-blocked mask $S_1$. Second, we remove filters based on entries' signs through a mask $S_2$. Third, we remove filters according to the MRSS result through a mask $S_3$. Masks $S_2$ and $S_3$ can be combined into a single mask $S_2 \odot S_3$ that only removes filters: overall the pruning process consists of a $2n$-channel-blocked mask and filter removal, which justifies the statement of Theorem 2.

**Overview of the proof of Theorem 2**

We iteratively apply Lemma 12 to each layer while carefully controlling the approximation error via tools such as the Lipschitz property of ReLU and the Tensor Convolution Inequality (Lemma 20). More precisely, we show that (i) the approximation error does not increase too much at each layer; and (ii) all layer approximations can be combined to approximate the entire target network. Notice that Lemma 12 guarantees that, with high probability, we can approximate all possible target filters $K$ (with $\|K\|_1 \leq 1$) from the same 2-layered network. Hence, we get the result for the supremum over all possible choices of target filters.

## 5 Limitations and future work

In previous works [da Cunha et al., 2022, Burkholz, 2022a] the assumption that the kernel of every second layer has shape $1 \times 1 \times \ldots$ is only an artifact of the proof since one can readily prune entries of an arbitrarily shaped tensor to enforce the desired shape. In our case, however, the concept of structured pruning can be quite broad, and such reshaping via pruning might not fit some sparsity patterns, depending on the context. The hypothesis on the shape can be a relevant limitation for such use cases. The constructions proposed by Burkholz [2022a,b] appear as a promising direction to overcome this limitation, with the added benefit of reducing the depth overhead.

The convolution operation commonly employed in CNNs can be cumbersome at many points of our analysis. Exploring different concepts of convolution can be an interesting path for future work as it could lead to tidier proofs and more general results. For instance, employing a 3D convolution would spare a factor $c$ in Theorem 2.

Another limitation of our results is the restriction to ReLU as the activation function. Many previous works on the SLTH exploit the fact that ReLU satisfies the identity $x = \phi(x) - \phi(-x)$. Burkholz [2022a] leveraged that to obtain an SLTH result for CNNs with activation functions $f$ for which $f(x) - f(-x) \approx x$ around the origin. Our analysis, on the other hand, does not rely on such property, so adapting the approach of [Burkholz, 2022a] to our setting is not straightforward.

Finally, we remark that the assumption of normally distributed weights might be relaxed. Borst et al. [2022] provided an MRSSP result for independent random variables whose distribution converges "fast enough" to a Gaussian one.[4] We believe our arguments can serve well as baselines to generalise our results to support random weights distributed as such.

## Acknowledgments and Disclosure of Funding

Francesco d'Amore is supported by the MUR Fare2020 Project "PAReCoDi" CUP J43C22000970001.

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

# Appendix

## A  Technical tools

### A.1  Concentration inequalities

**Lemma 14** (Most-probable normal interval). *Let $X$ follow a zero-mean normal distribution with variance $\phi^2$. For any $z, \varepsilon \in \mathbb{R}$*

$$\Pr\left(X \in [z - \varepsilon, z + \varepsilon]\right) \leq \Pr\left(X \in [-\varepsilon, \varepsilon]\right).$$

*Proof.* Let $\varphi(x)$ denote the probability density function of $X$. Then,

$$\Pr\left(X \in [-\varepsilon, \varepsilon]\right) - \Pr\left(X \in [z - \varepsilon, z + \varepsilon]\right) = \int_{-\varepsilon}^{\varepsilon} \varphi(x)\, \mathrm{d}x - \int_{z-\varepsilon}^{z+\varepsilon} \varphi(x)\, \mathrm{d}x.$$

If $z - \varepsilon \geq \varepsilon$ or $z + \varepsilon \leq -\varepsilon$, the thesis is trivial as $\varphi(|x|)$ decreases in $x$. W.l.o.g., suppose $z$ is positive and $z - \varepsilon < \varepsilon$. Then, $-\varepsilon < z - \varepsilon < \varepsilon < z + \varepsilon$. It follows that

$$\int_{-\varepsilon}^{\varepsilon} \varphi(x)\, \mathrm{d}x - \int_{z-\varepsilon}^{z+\varepsilon} \varphi(x)\, \mathrm{d}x = \int_{-\varepsilon}^{z-\varepsilon} \varphi(x)\, \mathrm{d}x - \int_{\varepsilon}^{z+\varepsilon} \varphi(x)\, \mathrm{d}x$$

$$= \int_{-\varepsilon}^{z-\varepsilon} \varphi(x) - \varphi(x + 2\varepsilon)\, \mathrm{d}x$$

which is non-negative as $\varphi(x) \geq \varphi(x + 2\varepsilon)$ for $x \geq -\varepsilon$. $\qquad\square$

**Lemma 15** (Second moment method). *If $Z$ is a non-negative random variable then*

$$\Pr\left(Z > 0\right) \geq \frac{\mathbb{E}\left[Z\right]^2}{\mathbb{E}\left[Z^2\right]}.$$

**Lemma 16** (Chernoff-Hoeffding bounds [Dubhashi and Panconesi, 2009]). *Let $X_1, X_2, \ldots, X_n$ be independent random variables such that $\Pr\left(0 \leq X_i \leq 1\right) = 1$ for all $i \in [n]$. Let $X = \sum_{i=1}^{n} X_i$ and $\mathbb{E}[X] = \mu$. Then, for any $\delta \in (0, 1)$ the following holds:*

1. *if $\mu \leq \mu_+$, then $\Pr\left(X \geq (1 + \delta)\mu_+\right) \leq \exp\left(-\frac{\delta^2 \mu_+}{3}\right)$;*

2. *if $0 \leq \mu_- \leq \mu$, then $\Pr\left(X \leq (1 - \delta)\mu_-\right) \leq \exp\left(-\frac{\delta^2 \mu_+}{2}\right)$.*

**Lemma 17** (Corollary of [Laurent and Massart, 2000, Lemma 1]). *Let $X \sim \chi_d^2$ be a chi-squared random variable with $d$ degrees of freedom. For any $t > 0$, it holds that*

1. $\Pr\left(X \geq d + 2\sqrt{dt} + 2t\right) \leq \exp\left(-t\right)$;

2. $\Pr\left(X \leq d - 2\sqrt{dt}\right) \leq \exp\left(-t\right)$.

### A.2  Supporting results

**Lemma 18** (NSN with positive scalar). *If a $d$-dimensional random vector $Y$ is such that, for each $i \in [d]$, $Y_i = \tilde{Z} \cdot \tilde{Z}_i$, where $\tilde{Z}_1, \ldots, \tilde{Z}_n$ are identically distributed random variables following a standard normal distribution, $\tilde{Z}$ is a half-normal distribution,[5] and $\tilde{Z}, \tilde{Z}_1, \ldots, \tilde{Z}_n$ are independent, then $Y$ follows an NSN distribution.*

*Proof.* By Definition 4, $Y$ is NSN if, for each $i \in [d]$, $Y_i = Z \cdot Z_i$ where $Z, Z_1, \ldots, Z_n$ are i.i.d. random variables following a standard normal distribution. We know that $\tilde{Z} = |Z|$, therefore $\tilde{Z}_i = \text{sign}\left(Z\right) \text{sign}\left(Z_i\right) |Z_i|$ for each $i = 1, \ldots, n$, where $Z, Z_1, \ldots, Z_n$ are i.i.d. standard normal

---

[5]I.e. $\tilde{Z} = |Z|$ where $Z$ is a standard normal distribution.

random variables, as $\operatorname{sign}(Z)\operatorname{sign}(Z_i)$ is independent of $\operatorname{sign}(Z)$ and of $\operatorname{sign}(Z)\operatorname{sign}(Z_j)$ for $i \neq j$. Then,

$$
\begin{aligned}
Y_i &= \tilde{Z} \cdot \tilde{Z}_i \\
&= |Z| \cdot \operatorname{sign}(Z)\operatorname{sign}(Z_i)|Z_i| \\
&= \operatorname{sign}(Z)|Z| \cdot \operatorname{sign}(Z_i)|Z_i| \\
&= Z \cdot Z_i,
\end{aligned}
$$

implying the thesis.

$\square$

**Corollary 19** (of Theorem 5). *Let $d, k,$ and $n$ be positive integers with $n \geq C_1 dk^2 \log\left(\frac{1}{\varepsilon}\right)$ and $k \geq C_2 d^3 \log \frac{d}{\varepsilon}$ for some universal constants $C_1, C_2 \in \mathbb{R}_{>0}$. Let $X_1, \ldots, X_n$ be d-dimensional i.i.d. NSN random vectors. For any $0 < \varepsilon \leq \frac{1}{4}$ it holds*

$$
\Pr\left( \forall \vec{z} \in \mathbb{R}^d : \|\vec{z}\|_1 \leq 1, \exists S : |S| = k, \left\| \left(\sum_{i \in S} X_i \right) - \vec{z} \right\|_{\max} \leq \varepsilon \right) \geq 1 - \varepsilon.
$$

*Proof.* Observe that the set $[-1, 1]^d$ can be "partitioned" in $1/\varepsilon^d$ many $\infty$-norm balls of radius $\varepsilon$. Let $s = \left\lceil C_1 d \log\left(\frac{1}{\varepsilon}\right) \right\rceil$ and let us partition the $n$ vectors $X_1, \ldots, X_n$ in $s$ disjoint sets $G_1, \ldots, G_s$ of at least $k^2$ vectors each. By Theorem 5, there is a constant $c \in (0, 1)$ such that for each group $G_i$ $(i \in [s])$

$$
\Pr\left( \exists S \subset G_i : |S| = k, \left\| \left(\sum_{i \in S} X_i \right) - \vec{z} \right\|_{\max} \leq \varepsilon \right) \geq c. \tag{3}
$$

It follows that

$$
\Pr\left( \forall \vec{z} \in \mathbb{R}^d : \|\vec{z}\|_1 \leq 1, \exists S : |S| = k, \left\| \left(\sum_{i \in S} X_i \right) - \vec{z} \right\|_{\max} \leq \varepsilon \right)
$$

$$
\geq \Pr\left( \forall \vec{z} \in \mathbb{R}^d : \|\vec{z}\|_1 \leq 1, \exists i \in [s], \exists S \subset G_i : |S| = k, \left\| \left(\sum_{i \in S} X_i \right) - \vec{z} \right\|_{\max} \leq \varepsilon \right)
$$

$$
= 1 - \Pr\left( \exists \vec{z} \in \mathbb{R}^d : \|\vec{z}\|_1 \leq 1, \forall i \in [s], \forall S \subset G_i : |S| = k, \left\| \left(\sum_{i \in S} X_i \right) - \vec{z} \right\|_{\max} > \varepsilon \right)
$$

$$
\geq 1 - \frac{1}{\varepsilon^d} \Pr\left( \forall i \in [s], \forall S \subset G_i : |S| = k, \left\| \left(\sum_{i \in S} X_i \right) - \vec{z} \right\|_{\max} > \varepsilon \right) \tag{4}
$$

$$
\geq 1 - \frac{1}{\varepsilon^d} (1 - c)^{\left\lceil C_1 d \log\left(\frac{1}{\varepsilon}\right) \right\rceil}, \tag{5}
$$

where Eq. 4 holds by the union bound, Eq. 5 comes from Eq. 3 and the independence of the variables across different $G_i$. By choosing $C_1$ large enough,

$$
1 - \varepsilon^{-d} (1 - c)^{\left\lceil C_1 d \log\left(\frac{1}{\varepsilon}\right) \right\rceil} = 1 - 2^{d \log \frac{1}{\varepsilon} - \left\lceil C_1 d \log\left(\frac{1}{\varepsilon}\right) \right\rceil \log \frac{1}{1-c}} \geq 1 - \varepsilon.
$$

$\square$

**Lemma 20** (Tensor Convolution Inequality). *Given real tensors $K$ and $X$ of respective sizes $d \times d' \times c_0 \times c_1$ and $D \times D' \times c_0$, it holds*

$$
\|K * X\|_{\max} \leq \|K\|_1 \cdot \|X\|_{\max}.
$$

*Proof.* We have

$$
\|K * X\|_{\max}
$$

$$\leq \max_{i,j\in[D],\ell\in[c_1]} \sum_{i',j'\in[d],k\in[c]} |K_{i',j',k,\ell} X_{i-i'+1,j-j'+1,k}|$$

$$\leq \max_{i,j\in[D],\ell\in[c_1]} \left( \sum_{i',j'\in[d],k\in[c]} |K_{i',j',k,\ell}| \right) \|X\|_{\max}$$

$$\leq \max_{i,j\in[D],\ell\in[c_1]} \|K\|_1 \cdot \|X\|_{\max}$$

$$= \|K\|_1 \cdot \|X\|_{\max}.$$

$\square$

# B  Omitted proofs and results

## B.1  Multidimensional Random Subset Sum for normally-scaled normal vectors

*Proof of Lemma 7.* By Definition 4, the $j$-th entry of each vector $X_i$ is $(X_i)_j = Z_i \cdot Z_{i,j}$ where each $Z_i$ and $Z_{i,j}$ are i.i.d. random variables following a standard normal distribution. Let $\mathcal{E}^{(\ddagger)}$ be the event that $k\left(1 - 2\sqrt{c_d}\right) \leq \sum_{i=1}^k Z_i^2 \leq k\left(1 + 2\sqrt{c_d} + 2c_d\right)$, and denote $X = \sum_{i=1}^k X_i$. By the law of total probability, it holds that

$$\Pr\left(X \in I_\varepsilon\left(\vec{z}\right)\right) = \mathbb{E}_{Z_1,\ldots,Z_n}\left[\Pr\left(X \in I_\varepsilon\left(\vec{z}\right) \mid Z_1,\ldots,Z_k\right)\right].$$

As, conditional on $Z_1,\ldots,Z_k$, the $d$ entries of $X$ are independent, it follows that

$$\mathbb{E}_{Z_1,\ldots,Z_n}\left[\Pr\left(X \in I_\varepsilon\left(\vec{z}\right) \mid Z_1,\ldots,Z_k\right)\right]$$

$$= \mathbb{E}_{Z_1,\ldots,Z_n}\left[\prod_{j=1}^d \Pr\left((X)_j \in I_\varepsilon\left(z_i\right) \mid Z_1,\ldots,Z_k\right)\right]$$

$$\geq \mathbb{E}_{Z_1,\ldots,Z_n}\left[\prod_{j=1}^d \Pr\left((X)_j \in I_\varepsilon\left(z_i\right) \mid Z_1,\ldots,Z_k, \mathcal{E}^{(\ddagger)}\right)\right] \Pr\left(\mathcal{E}^{(\ddagger)}\right), \qquad (6)$$

where the inequality in Eq. 6 holds by applying again the law of total probability.

Conditional on $Z_1,\ldots,Z_k$, we have that $(X)_j \sim \mathcal{N}(0, \sum_{i=1}^k Z_i^2)$. Hence,

$$\mathbb{E}_{Z_1,\ldots,Z_n}\left[\prod_{j=1}^d \Pr\left((X)_j \in I_\varepsilon\left(z_i\right) \mid Z_1,\ldots,Z_k, \mathcal{E}^{(\ddagger)}\right)\right] \Pr\left(\mathcal{E}^{(\ddagger)}\right)$$

$$\geq \mathbb{E}_{Z_1,\ldots,Z_k}\left[\prod_{j=1}^d \left(\frac{2\varepsilon}{\sqrt{\pi\left(\sum_{i=1}^k Z_i^2\right)}} \exp\left(-\frac{(|z_i|+\varepsilon)^2}{2\sum_{i=1}^k Z_i^2}\right)\right)\bigg| \mathcal{E}^{(\ddagger)}\right]$$

$$\cdot \Pr\left(\mathcal{E}^{(\ddagger)}\right)$$

Notice that the term $\sum_i Z_i^2$ is a sum of chi-square random variables, for which there are known concentration bounds (Lemma 17). By definition of $\mathcal{E}^{(\ddagger)}$ and by applying Lemma 17 to estimate the term $\Pr\left(\mathcal{E}^{(\ddagger)}\right)$, we get that

$$\mathbb{E}_{Z_1,\ldots,Z_k}\left[\prod_{j=1}^d \left(\frac{2\varepsilon}{\sqrt{\pi\left(\sum_{i=1}^k Z_i^2\right)}} \exp\left(-\frac{(|z_i|+\varepsilon)^2}{2\sum_{i=1}^k Z_i^2}\right)\right)\bigg| \mathcal{E}^{(\ddagger)}\right]$$

$$\cdot \Pr\left(\mathcal{E}^{(\ddagger)}\right)$$

$$\geq \left(\frac{2\varepsilon}{\sqrt{\pi\left(1+2\sqrt{c_d}+2c_d\right)k}}\right)^d \exp\left(-\frac{\sum_i |z_i|^2 + 2\varepsilon\sum_i |z_i| + d\varepsilon^2}{2\left(1-2\sqrt{c_d}\right)k}\right) \Pr\left(\mathcal{E}^{(\updownarrow)}\right)$$

$$= \left(\frac{2\varepsilon}{\sqrt{\pi\left(1+2\sqrt{c_d}+2c_d\right)k}}\right)^d \exp\left(-\frac{\|\vec{z}\|_2^2 + 2\varepsilon\|\vec{z}\|_1 + d\varepsilon^2}{2\left(1-2\sqrt{c_d}\right)k}\right) \Pr\left(\mathcal{E}^{(\updownarrow)}\right)$$

$$\geq \left(\frac{2\varepsilon}{\sqrt{\pi\left(1+2\sqrt{c_d}+2c_d\right)k}}\right)^d \exp\left(-\frac{\|\vec{z}\|_2^2 + 2\varepsilon\|\vec{z}\|_1 + d\varepsilon^2}{2\left(1-2\sqrt{c_d}\right)k}\right)\left(1-2e^{-c_d k}\right).$$

As $c_d k \geq 1$ by hypotheses, $1 - 2e^{-c_d k} \geq 1/4$. Then,

$$\left(\frac{2\varepsilon}{\sqrt{\pi\left(1+2\sqrt{c_d}+2c_d\right)k}}\right)^d \exp\left(-\frac{\|\vec{z}\|_2^2 + 2\varepsilon\|\vec{z}\|_1 + d\varepsilon^2}{2\left(1-2\sqrt{c_d}\right)k}\right)\left(1-2e^{-c_d k}\right)$$

$$\geq \frac{1}{4}\left(\frac{2\varepsilon}{\sqrt{\pi\left(1+2\sqrt{c_d}+2c_d\right)k}}\right)^d \exp\left(-\frac{\|\vec{z}\|_2^2 + 2\varepsilon\|\vec{z}\|_1 + d\varepsilon^2}{2\left(1-2\sqrt{c_d}\right)k}\right)$$

$$\geq \frac{1}{4}\left(\frac{2\varepsilon}{\sqrt{\pi\left(1+2\sqrt{c_d}+2c_d\right)k}}\right)^d \exp\left(-\frac{k + 2\varepsilon\sqrt{k} + d\varepsilon^2}{2\left(1-2\sqrt{c_d}\right)k}\right) \tag{7}$$

$$= \frac{1}{4}\left(\frac{2\varepsilon}{\sqrt{\pi\left(1+2\sqrt{c_d}+2c_d\right)k}}\right)^d \exp\left(-\frac{1 + \frac{2\varepsilon}{\sqrt{k}} + \frac{d\varepsilon^2}{k}}{2\left(1-2\sqrt{c_d}\right)}\right)$$

$$\geq \frac{1}{4}\left(\frac{2\varepsilon}{\sqrt{\pi\left(1+2\sqrt{c_d}+2c_d\right)k}}\right)^d \exp\left(-\frac{1 + \frac{1}{8} + \frac{1}{16}}{2\left(1-2\sqrt{c_d}\right)}\right) \tag{8}$$

$$\geq \frac{1}{16}\left(\frac{2\varepsilon}{\sqrt{\pi\left(1+2\sqrt{c_d}+2c_d\right)k}}\right)^d, \tag{9}$$

where we have used that $\|\vec{z}\|_2 \leq \|\vec{z}\|_1 \leq \sqrt{k}$ in Ineq. 7, that $k \geq 16$, $k \geq d$, and $\varepsilon < 1/4$ in Ineq. 8, and that

$$\exp\left(-\frac{1 + \frac{1}{8} + \frac{1}{16}}{2\left(1-2\sqrt{c_d}\right)}\right) \geq \exp\left(-\frac{1 + \frac{1}{8} + \frac{1}{16}}{2\left(1-2\sqrt{\frac{1}{16}}\right)}\right) \geq \frac{1}{16}$$

in Ineq. 9. $\qquad\square$

*Proof of Lemma 8.* Let $n = k + j$. Since the $X_i$s are NSN random vectors, for each $i \in [n]$ and $j \in [d]$ we can write the $j$-th entry of $X_i$ as $(X_i)_j = Z_i \cdot Z_{i,j}$ where the variables in $\{Z_i\}_{i\in[n]}$ and in $\{Z_{i,j}\}_{i\in[n],j\in[d]}$ are i.i.d. random variables following a standard normal distribution. By the law of total probability, we have

$$\Pr\left(A + B \in I_\varepsilon\left(\vec{z}\right), B + C \in I_\varepsilon\left(\vec{z}\right)\right)$$
$$= \mathbb{E}_{Z_1,\ldots,Z_n}\left[\Pr\left(A + B \in I_\varepsilon\left(\vec{z}\right), B + C \in I_\varepsilon\left(\vec{z}\right) \mid Z_1,\ldots,Z_n\right)\right]$$
$$= \mathbb{E}_{Z_1,\ldots,Z_n}\left[\prod_{i=1}^d \Pr\left(A_i + B_i \in I_\varepsilon\left(z_i\right), B_i + C_i \in I_\varepsilon\left(z_i\right) \mid Z_1,\ldots,Z_n\right)\right], \tag{10}$$

where the latter equality holds by independence.

Then,

$$\Pr\left(A_i + B_i \in I_\varepsilon\left(z_i\right), B_i + C_i \in I_\varepsilon\left(z_i\right) \mid Z_1, \ldots, Z_n\right)$$
$$= \mathbb{E}_{B_i}\left[\Pr\left(A_i \in I_\varepsilon\left(z_i - B_i\right), C_i \in I_\varepsilon\left(z_i - B_i\right) \mid Z_1, \ldots, Z_n, B_i\right)\right]$$
$$= \mathbb{E}_{B_i}\left[\Pr\left(A_i \in I_\varepsilon\left(z_i - B_i\right) \mid Z_1, \ldots, Z_n, B_i\right)\Pr\left(C_i \in I_\varepsilon\left(z_i - B_i\right) \mid Z_1, \ldots, Z_n, B_i\right)\right],$$

where the latter inequality holds by independence of $A_i$ and $C_i$. By Lemma 14, it holds that

$$\mathbb{E}_{B_i}\left[\Pr\left(A_i \in I_\varepsilon\left(z_i - B_i\right) \mid Z_1, \ldots, Z_n, B_i\right)\Pr\left(C_i \in I_\varepsilon\left(z_i - B_i\right) \mid Z_1, \ldots, Z_n, B_i\right)\right]$$
$$\leq \mathbb{E}_{B_i}\left[\Pr\left(A_i \in I_\varepsilon\left(0\right) \mid Z_1, \ldots, Z_n, B_i\right)\Pr\left(C_i \in I_\varepsilon\left(0\right) \mid Z_1, \ldots, Z_n, B_i\right)\right]$$
$$= \Pr\left(A_i \in I_\varepsilon\left(0\right) \mid Z_1, \ldots, Z_n\right)\Pr\left(C_i \in I_\varepsilon\left(0\right) \mid Z_1, \ldots, Z_n\right)$$
$$\leq \frac{2\varepsilon}{\sqrt{\pi\left(\sum_{r=1}^{j} Z_r^2\right)}} \cdot \frac{2\varepsilon}{\sqrt{\pi\left(\sum_{r=k+1}^{k+j} Z_r^2\right)}}, \tag{11}$$

where the latter inequality comes from the fact that, conditional on $Z_1, \ldots, Z_n$, we have that $A_i \sim \mathcal{N}(0, \sum_{r=1}^{j} Z_r^2)$, $B_i \sim \mathcal{N}(0, \sum_{r=j+1}^{k} Z_r^2)$, and $C_i \sim \mathcal{N}(0, \sum_{r=k+1}^{k+j} Z_r^2)$ for each $i \in [d]$.

We now proceed similarly to the proof of Lemma 7. We denote the event that $\left(1 - 2\sqrt{c_d}\right)j \leq \sum_{i=1}^{j} Z_i^2, \sum_{i=k+1}^{k+j} Z_i^2$ by $\mathcal{E}^{(\downarrow)}$. Then, by Eq. 10 and the law of total probability, we have that

$$\Pr\left(A + B \in I_\varepsilon\left(\vec{z}\right), B + C \in I_\varepsilon\left(\vec{z}\right)\right)$$
$$= \mathbb{E}_{Z_1, \ldots, Z_n}\left[\prod_{i=1}^{d} \Pr\left(A_i + B_i \in I_\varepsilon\left(z_i\right), B_i + C_i \in I_\varepsilon\left(z_i\right) \mid Z_1, \ldots, Z_n\right)\right]$$
$$\leq \mathbb{E}_{Z_1, \ldots, Z_n}\left[\prod_{i=1}^{d} \Pr\left(A_i + B_i, B_i + C_i \in I_\varepsilon\left(z_i\right) \mid Z_1, \ldots, Z_n, \mathcal{E}^{(\downarrow)}\right)\right] + \Pr\left(\overline{\mathcal{E}^{(\downarrow)}}\right).$$

Eq. 11 implies that

$$\mathbb{E}_{Z_1, \ldots, Z_n}\left[\prod_{i=1}^{d} \Pr\left(A_i + B_i, B_i + C_i \in I_\varepsilon\left(z_i\right) \mid Z_1, \ldots, Z_n, \mathcal{E}^{(\downarrow)}\right)\right] + \Pr\left(\overline{\mathcal{E}^{(\downarrow)}}\right)$$
$$\leq \mathbb{E}_{Z_1, \ldots, Z_n}\left[\prod_{i=1}^{d} \frac{2\varepsilon}{\sqrt{\pi\left(\sum_{r=1}^{j} Z_r^2\right)}} \cdot \frac{2\varepsilon}{\sqrt{\pi\left(\sum_{r=k+1}^{k+j} Z_r^2\right)}} \middle| \mathcal{E}^{(\downarrow)}\right] + \Pr\left(\overline{\mathcal{E}^{(\downarrow)}}\right)$$
$$= \mathbb{E}_{Z_1, \ldots, Z_n}\left[\left(\frac{4\varepsilon^2}{\pi\sqrt{\left(\sum_{r=1}^{j} Z_r^2\right)\left(\sum_{r=k+1}^{k+j} Z_r^2\right)}}\right)^{d} \middle| \mathcal{E}^{(\downarrow)}\right] + \Pr\left(\overline{\mathcal{E}^{(\downarrow)}}\right).$$

By independence of $\sum_{r=1}^{j} Z_r^2$ and $\sum_{r=k+1}^{k+j} Z_r^2$ and by Lemma 17, we obtain that

$$\mathbb{E}_{Z_1, \ldots, Z_n}\left[\left(\frac{4\varepsilon^2}{\pi\sqrt{\left(\sum_{i=1}^{j} Z_i^2\right)\left(\sum_{i=k+1}^{k+j} Z_i^2\right)}}\right)^{d} \middle| \mathcal{E}^{(\downarrow)}\right] + \Pr\left(\overline{\mathcal{E}^{(\downarrow)}}\right)$$
$$\leq \left(\frac{4\varepsilon^2}{\pi j\left(1 - 2\sqrt{c_d}\right)}\right)^{d} + \Pr\left(\overline{\mathcal{E}^{(\downarrow)}}\right)$$
$$\leq \left(\frac{4\varepsilon^2}{\pi j\left(1 - 2\sqrt{c_d}\right)}\right)^{d} + 2\exp\left(-c_d j\right)$$

$$= \exp\left(-d\log\frac{\pi j\left(1-2\sqrt{c_d}\right)}{4\varepsilon^2}\right) + 2\exp\left(-c_d k\right).$$

Finally, for a large enough constant $C$, the hypothesis on $k$ implies that $k \geq 2\frac{d}{c_d}\log\frac{\pi k\left(1-2\sqrt{c_d}\right)}{4\varepsilon^2}$. Hence,

$$\exp\left(-d\log\frac{\pi j\left(1-2\sqrt{c_d}\right)}{4\varepsilon^2}\right) + 2\exp\left(-c_d k\right) \leq 3\left(\frac{4\varepsilon^2}{\pi\left(1-2\sqrt{c_d}\right)k}\right)^d.$$

$\square$

*Proof of Theorem 5.* We use the second moment method (Lemma 15) on the $\varepsilon$-subset-sum number $T_{n,k}$ of $X_1, \ldots, X_n$. Thus, we aim to provide a lower bound on the right-hand side of

$$\Pr\left(T > 0\right) \geq \frac{\mathbb{E}[T_{n,k}]^2}{\mathbb{E}[T_{n,k}^2]}. \tag{12}$$

Equivalently, we can provide an upper bound on the inverse $\frac{\mathbb{E}[T_{n,k}^2]}{\mathbb{E}[T_{n,k}]^2}$. By Lemma 9

$$\mathbb{E}\left[T_{n,k}^2\right] = \binom{n}{k}^2 \sum_{j=0}^{k} \Pr\left(|\mathrm{S}\cap\mathrm{S}'| = k - j\right)\Pr\left(\mathcal{E}_{S_0}^{(\vec{z})}\cap\mathcal{E}_{S_j}^{(\vec{z})}\right) \tag{13}$$

where $\mathrm{S}, \mathrm{S}', S_i$ and $\mathcal{E}_S^{(\vec{z})}$ are defined as in the statement of the lemma. Observe also that

$$\mathbb{E}\left[T_{n,k}\right] = \sum_{S\in\binom{[n]}{k}} \mathbb{E}\left[\mathbf{1}_{\mathcal{E}_S^{(\vec{z})}}\right] = \sum_{S\in\binom{[n]}{k}} \Pr\left(\mathcal{E}_S^{(\vec{z})}\right) = \binom{n}{k}\Pr\left(\mathcal{E}_{S_0}^{(\vec{z})}\right). \tag{14}$$

By Eq.s 13 and 14, we have

$$\frac{\mathbb{E}\left[T_{n,k}^2\right]}{\mathbb{E}\left[T_{n,k}\right]^2} = \frac{\binom{n}{k}^2}{\mathbb{E}\left[T_{n,k}\right]^2} \sum_{j=0}^{k} \Pr\left(|\mathrm{S}\cap\mathrm{S}'| = k - j\right)\Pr\left(\mathcal{E}_{S_0}^{(\vec{z})}\cap\mathcal{E}_{S_j}^{(\vec{z})}\right)$$

$$= \sum_{j=0}^{k} \Pr\left(|\mathrm{S}\cap\mathrm{S}'| = k - j\right)\frac{\Pr\left(\mathcal{E}_{S_0}^{(\vec{z})}\cap\mathcal{E}_{S_j}^{(\vec{z})}\right)}{\Pr\left(\mathcal{E}_{S_0}^{(\vec{z})}\right)^2}. \tag{15}$$

As for the denominator of the second term in Eq. 15, by Lemma 7 we have

$$\Pr\left(\mathcal{E}_{S_0}^{(\vec{z})}\right)^2 \geq \frac{1}{256}\left(\frac{4\varepsilon^2}{\pi\left(1+2\sqrt{c_d}+2c_d\right)k}\right)^d. \tag{16}$$

As for the numerator of the second term in Eq. 15, Lemma 8 implies that we have

$$\Pr\left(\mathcal{E}_{S_0}^{(\vec{z})}\cap\mathcal{E}_{S_j}^{(\vec{z})}\right) \leq 3\left(\frac{4\varepsilon^2}{\pi\left(1-2\sqrt{c_d}\right)j}\right)^d. \tag{17}$$

By plugging Eq. 17 and Eq. 16 in Eq. 15, for $j \geq k\left(1-\frac{1}{d}\right)$ and $d > 1$ we can upper bound the factor $\frac{\Pr\left(\mathcal{E}_{S_0}^{(\vec{z})}\cap\mathcal{E}_{S_j}^{(\vec{z})}\right)}{\Pr\left(\mathcal{E}_{S_0}^{(\vec{z})}\right)^2}$ of the summation as follows:

$$\frac{\Pr\left(\mathcal{E}_{S_0}^{(\vec{z})}\cap\mathcal{E}_{S_j}^{(\vec{z})}\right)}{\Pr\left(\mathcal{E}_{S_0}^{(\vec{z})}\right)^2} \leq \frac{3\left(\frac{4\varepsilon^2}{\pi\left(1-2\sqrt{c_d}\right)j}\right)^d}{\frac{1}{256}\left(\frac{4\varepsilon^2}{\pi\left(1+2\sqrt{c_d}+2c_d\right)k}\right)^d}$$

$$= 768 \left( \frac{\left(1 + 2\sqrt{c_d} + 2c_d\right) k}{\left(1 - 2\sqrt{c_d}\right) j} \right)^d.$$

As $j \geq k\left(1 - \frac{1}{d}\right)$ with $d > 1$, then

$$768 \left( \frac{\left(1 + 2\sqrt{c_d} + 2c_d\right) k}{\left(1 - 2\sqrt{c_d}\right) j} \right)^d \leq 768 \left( \frac{\left(1 + 2\sqrt{c_d} + 2c_d\right)}{\left(1 - 2\sqrt{c_d}\right)\left(1 - \frac{1}{d}\right)} \right)^d$$

$$\leq 768 \left( \frac{2 + 7\sqrt{c_d}}{2 - 7\sqrt{c_d}} \right)^d$$

$$\leq 270801, \tag{18}$$

because $\left( \frac{\left(1 + 2\sqrt{c_d} + 2c_d\right)}{\left(1 - 2\sqrt{c_d}\right)\left(1 - \frac{1}{d}\right)} \right)^d$ is maximised for $d = 4$. Let $C' = 270801$. If $d > 1$, by plugging Eq. 18 in Eq. 15, we have that

$$\frac{\mathbb{E}\left[T_{n,k}^2\right]}{\mathbb{E}\left[T_{n,k}\right]^2} = \sum_{j=0}^{\lceil k - \frac{k}{d} \rceil - 1} \Pr\left(|S \cap S'| = k - j\right) \frac{\Pr\left(\mathcal{E}_{S_0}^{(\vec{z})} \cap \mathcal{E}_{S_j}^{(\vec{z})}\right)}{\Pr\left(\mathcal{E}_{S_0}^{(\vec{z})}\right)^2}$$

$$+ \sum_{j=\lceil k - \frac{k}{d} \rceil}^{k} \Pr\left(|S \cap S'| = k - j\right) \frac{\Pr\left(\mathcal{E}_{S_0}^{(\vec{z})} \cap \mathcal{E}_{S_j}^{(\vec{z})}\right)}{\Pr\left(\mathcal{E}_{S_0}^{(\vec{z})}\right)^2}$$

$$\leq \sum_{j=0}^{\lceil k - \frac{k}{d} \rceil - 1} \Pr\left(|S \cap S'| = k - j\right) \frac{\Pr\left(\mathcal{E}_{S_0}^{(\vec{z})} \cap \mathcal{E}_{S_j}^{(\vec{z})}\right)}{\Pr\left(\mathcal{E}_{S_0}^{(\vec{z})}\right)^2} + C'.$$

As $\Pr\left(\mathcal{E}_{S_0}^{(\vec{z})} \cap \mathcal{E}_{S_j}^{(\vec{z})}\right) \leq \Pr\left(\mathcal{E}_{S_0}^{(\vec{z})}\right)$, then

$$\sum_{j=0}^{\lceil k - \frac{k}{d} \rceil - 1} \Pr\left(|S \cap S'| = k - j\right) \frac{\Pr\left(\mathcal{E}_{S_0}^{(\vec{z})} \cap \mathcal{E}_{S_j}^{(\vec{z})}\right)}{\Pr\left(\mathcal{E}_{S_0}^{(\vec{z})}\right)^2} + C'$$

$$\leq \frac{1}{\Pr\left(\mathcal{E}_{S_0}^{(\vec{z})}\right)} \sum_{j=0}^{\lceil k - \frac{k}{d} \rceil - 1} \Pr\left(|S \cap S'| = k - j\right) + C'$$

$$\leq \frac{\Pr\left(|S \cap S'| > \frac{k}{d}\right)}{\Pr\left(\mathcal{E}_{S_0}^{(\vec{z})}\right)} + C'. \tag{19}$$

Notice that, if $d = 1$, the same bound holds as $\Pr\left(|S \cap S'| > \frac{k}{d}\right) = 0$. We now observe that, by the law of total probability

$$\Pr\left(|S \cap S'| \geq \frac{k}{d}\right) = \sum_{\tilde{S} \in \binom{[n]}{k}} \Pr\left(S = \tilde{S}\right) \Pr\left(|S \cap S'| \geq \frac{k}{d} \,\middle|\, S = \tilde{S}\right). \tag{20}$$

Conditional on $S = \tilde{S}$, $|S \cap S'|$ is a hypergeometric random variable with

$$\mathbb{E}\left[|S \cap S'| \,\middle|\, S = \tilde{S}\right] = \sum_{i \in \tilde{S}} \Pr\left(i \in S'\right) = k \Pr\left(1 \in S\right) = \frac{k^2}{n}.$$

Since $n \geq k^2$, then $\frac{k^2}{n} \leq 1$. Hence, since Chernoff bounds holds for the hypergeometric distribution [Doerr, 2020, Theorem 1.10.25]

$$\Pr\left(|S \cap S'| \geq \frac{k}{d} \,\bigg|\, S = \tilde{S}\right) \geq \Pr\left(|S \cap S'| \geq \frac{k^2}{n} + \left(\frac{k}{d} - 1\right) \,\bigg|\, S = \tilde{S}\right)$$

$$\leq \exp\left(-2\frac{\left(\frac{k}{d} - 1\right)^2}{k}\right)$$

$$\leq \exp\left(-2\frac{k}{d^2}\left(1 - \frac{d}{k}\right)^2\right). \tag{21}$$

Substituting Eq. 21 in Eq. 20 we get

$$\Pr\left(|S \cap S'| \geq \frac{k}{d}\right) \leq \exp\left(-2\frac{k}{d^2}\left(1 - \frac{d}{k}\right)^2\right). \tag{22}$$

We can now keep bounding from above $\frac{\mathbb{E}[T_{n,k}^2]}{\mathbb{E}[T_{n,k}]^2}$ by plugging Eq. 22 in Eq. 19:

$$\frac{\Pr\left(|S \cap S'| \geq \frac{k}{d}\right)}{\Pr\left(\mathcal{E}_{S_0}^{(\vec{z})}\right)} + C' \leq \frac{\exp\left(-2\frac{k}{d^2}\left(1 - \frac{d}{k}\right)^2\right)}{\Pr\left(\mathcal{E}_{S_0}^{(\vec{z})}\right)} + C'. \tag{23}$$

By Lemma 7, and since $1 + 2\sqrt{c_d} + 2c_d \leq 2$, we have

$$\frac{\exp\left(-2\frac{k}{d^2}\left(1 - \frac{d}{k}\right)^2\right)}{\Pr\left(\mathcal{E}_{S_0}^{(\vec{z})}\right)} + C'$$

$$\leq \frac{16\exp\left(-2\frac{k}{d^2}\left(1 - \frac{d}{k}\right)^2\right)}{\left(\frac{2\varepsilon}{\sqrt{\pi 2k}}\right)^d} + C'$$

$$= 16\exp\left(-2\frac{k}{d^2}\left(1 - \frac{d}{k}\right)^2 + d\log\left(\frac{\sqrt{\pi 2k}}{2\varepsilon}\right)\right) + C'. \tag{24}$$

By the hypothesis, since $k \geq Cd^3 \log\frac{d}{\varepsilon}$ for a large enough $C$, it holds that

$$16\exp\left(-2\frac{k}{d^2}\left(1 - \frac{d}{k}\right)^2 + d\log\left(\frac{\sqrt{\pi 2k}}{2\varepsilon}\right)\right) + C'$$

$$\leq C' + 16\exp\left(-d\log\frac{d}{\varepsilon}\right)$$

$$< C' + \frac{16}{e}, \tag{25}$$

where the latter inequality holds as $\varepsilon \leq 1/4$.

Plugging the inverse of the expression in Eq. 25 in Eq. 12 we obtain the thesis. $\qquad\square$

## B.2   Kernel Pruning

*Proof of Lemma 11.* $S_2 \in \{0,1\}^{\text{size}(V)}$ is such that $\tilde{V} = V \odot \tilde{S} = (V \odot S_1) \odot S_2$ contains only non-negative edges going from each input channel $t$ to the output channels $2(t-1)n+1, \ldots, (2t-1)n$, and only non-positive[6] edges going from each input channel $t$ to the output channels $(2t-1)n+1, \ldots, 2tn$, while all remaining edges are set to zero. Let us define some convenient notations before proceeding

---

[6]We consider 0 to be both non-negative and non-positive.

with the proofs. By $[n:m]$ we denote the set $\{n, n+1, \ldots, m\}$ for each pair of integers $n \le m \in \mathbb{N}$. In formulas, we obtain a tensor $\tilde{V}$ such that, for each $(t, k) \in [c] \times [2nc]$:

$$\left(V \odot \tilde{S}\right)_{1,1,t,k} = \begin{cases} V_{1,1,t,k} \cdot \mathbf{1}_{V_{1,1,t,k}>0} & \text{if } k \in [(2t-2)\,n+1 : (2t-1)\,n], \\ V_{1,1,t,k} \cdot \mathbf{1}_{V_{1,1,t,k}<0} & \text{if } k \in [(2t-1)\,n+1 : 2tn], \\ 0 & \text{otherwise.} \end{cases} \tag{26}$$

To simplify the notation, we define the following indicator functions: for any $(t, k) \in [c] \times [2nc]$,

$$\mathbf{1}_{\frac{k}{2n} \in \left(t-1, t-\frac{1}{2}\right]} = 1 \text{ iff } k \in [(2t-2)\,n+1 : (2t-1)\,n], \text{ and}$$

$$\mathbf{1}_{\frac{k}{2n} \in \left(t-\frac{1}{2}, t\right]} = 1 \text{ iff } k \in [(2t-1)\,n+1 : 2tn]. \tag{27}$$

For each $(i, j, k) \in [D] \times [D] \times [2nc]$, applying Eq. 26 and using Definition 10, it then holds

$$\left(\phi\left(\left(V \odot \tilde{S}\right) * X\right)\right)_{i,j,k}$$

$$= \phi\left(\sum_{t=1}^{c_0} \tilde{V}_{1,1,t,k} X_{i,j,t}\right)$$

$$= \phi\Bigg(\sum_{t=1}^{c_0} \Big(V_{1,1,t,k} X_{i,j,t} \cdot \mathbf{1}_{V_{1,1,t,k}>0} \mathbf{1}_{\frac{k}{2n} \in \left(t-1, t-\frac{1}{2}\right]}$$

$$+ V_{1,1,t,k} X_{i,j,t} \cdot \mathbf{1}_{V_{1,1,t,k}<0} \mathbf{1}_{\frac{k}{2n} \in \left(t-\frac{1}{2}, t\right]}\Big)\Bigg)$$

$$= \phi\left(\sum_{t=1}^{c_0} \left(V_{1,1,t,k}^+ X_{i,j,t} \mathbf{1}_{\frac{k}{2n} \in \left(t-1, t-\frac{1}{2}\right]} - V_{1,1,t,k}^- X_{i,j,t} \mathbf{1}_{\frac{k}{2n} \in \left(t-\frac{1}{2}, t\right]}\right)\right)$$

$$= \phi\Bigg(\sum_{t=1}^{c_0} \Big(V_{1,1,t,k}^+ (X_{i,j,t}^+ - X_{i,j,t}^-) \mathbf{1}_{\frac{k}{2n} \in \left(t-1, t-\frac{1}{2}\right]}$$

$$+ V_{1,1,t,k}^- (X_{i,j,t}^- - X_{i,j,t}^+) \mathbf{1}_{\frac{k}{2n} \in \left(t-\frac{1}{2}, t\right]}\Big)\Bigg). \tag{28}$$

Observe that only one term survives in the summation in Eq. 28, as there exists only one $t \in [c_0]$ such that $k \in [(2t-2)n+1 : 2tn]$, say $t^\star$. Moreover, out of the four additive terms in the expression

$$V_{1,1,t^\star,k}^+ (X_{i,j,t^\star}^+ - X_{i,j,t^\star}^-) \mathbf{1}_{\frac{k}{2n} \in \left(t^\star-1, t^\star-\frac{1}{2}\right]} + V_{1,1,t^\star,k}^- (X_{i,j,t^\star}^- - X_{i,j,t^\star}^+) \mathbf{1}_{\frac{k}{2n} \in \left(t^\star-\frac{1}{2}, t^\star\right]},$$

at most one is non-zero, due to Definition 10. The ReLU cancels out negative ones, implying that Eq. 28 can be rewritten without the ReLU as a sum of only non-negative terms (out of which, at most one is non-zero) as follows

$$\phi\Bigg(\sum_{t=1}^{c_0} \Big(V_{1,1,t,k}^+ (X_{i,j,t}^+ - X_{i,j,t}^-) \mathbf{1}_{\frac{k}{2n} \in \left(t-1, t-\frac{1}{2}\right]}$$

$$+ V_{1,1,t,k}^- (X_{i,j,t}^- - X_{i,j,t}^+) \mathbf{1}_{\frac{k}{2n} \in \left(t-\frac{1}{2}, t\right]}\Big)\Bigg)$$

$$= \sum_{t=1}^{c_0} \left(V_{1,1,t,k}^+ X_{i,j,t}^+ \mathbf{1}_{\frac{k}{2n} \in \left(t-1, t-\frac{1}{2}\right]} + V_{1,1,t,k}^- X_{i,j,t}^- \mathbf{1}_{\frac{k}{2n} \in \left(t-\frac{1}{2}, t\right]}\right). \tag{29}$$

Finally, by Eq. 26 and Eq. 27, $\tilde{V}_{1,1,t,k}^+ = 0$ if $\frac{k}{2n} \notin \left(t-1, t-\frac{1}{2}\right]$, and $\tilde{V}_{1,1,t,k}^- = 0$ if $\frac{k}{2n} \in \left(t-\frac{1}{2}, t\right]$, which means that in Eq. 29 we can ignore the indicator functions and further simplify the expression as

$$\sum_{t=1}^{c_0} \left(V_{1,1,t,k}^+ X_{i,j,t}^+ \mathbf{1}_{\frac{k}{2n} \in \left(t-1, t-\frac{1}{2}\right]} + V_{1,1,t,k}^- X_{i,j,t}^- \mathbf{1}_{\frac{k}{2n} \in \left(t-\frac{1}{2}, t\right]}\right)$$

$$= \sum_{t=1}^{c_0} \left(\tilde{V}_{1,1,t,k}^+ X_{i,j,t}^+ + \tilde{V}_{1,1,t,k}^- X_{i,j,t}^-\right)$$

$$= \left( \sum_{t=1}^{c_0} \tilde{V}_{1,1,t,k}^+ X_{i,j,t}^+ + \sum_{t=1}^{c_0} \tilde{V}_{1,1,t,k}^- X_{i,j,t}^- \right)$$

$$= \left( \tilde{V}^+ * X^+ + \tilde{V}^- * X^- \right)_{i,j,k}.$$

$\square$

*Proof of Lemma 12.* Adopting the same definitions as in Lemma 11 (and Eq. 26), for each $(r, s, t_1) \in [d] \times [d] \times [c_1]$ we have, by Lemma 11,

$$\left( U * \phi \left( \left( V \odot \tilde{S} \right) * X \right) \right)_{r,s,t_1}$$

$$= \left( U * \left( \left( \tilde{V}^+ * X^+ \right) + \left( \tilde{V}^- * X^- \right) \right) \right)_{r,s,t_1}$$

$$= \sum_{i,j\in[d],k\in[2nc_0]} U_{i,j,k,t_1} \cdot \left( \left( \tilde{V}^+ * X^+ \right) + \left( \tilde{V}^- * X^- \right) \right)_{r-i+1,s-j+1,k}$$

$$= \sum_{i,j\in[d],k\in[2nc_0]} U_{i,j,k,t_1} \cdot \sum_{t_0\in[c_0]} \left( \tilde{V}_{1,1,t_0,k}^+ \cdot X_{r-i+1,s-j+1,t_0}^+ \right.$$

$$\left. + \tilde{V}_{1,1,t_0,k}^- \cdot X_{r-i+1,s-j+1,t_0}^- \right)$$

$$= \sum_{t_0\in[c_0]} \sum_{i,j\in[d],k\in[2nc_0]} \left( U_{i,j,k,t_1} \cdot \tilde{V}_{1,1,t_0,k}^+ \right) \cdot X_{r-i+1,s-j+1,t_0}^+$$

$$+ \sum_{t_0\in[c_0]} \sum_{i,j\in[d],k\in[2nc_0]} \left( U_{i,j,k,t_1} \cdot \tilde{V}_{1,1,t_0,k}^- \right) \cdot X_{r-i+1,s-j+1,t_0}^-$$

$$= \sum_{i,j\in[d],t_0\in[c_0]} \left( \sum_{k\in[2nc_0]} U_{i,j,k,t_1} \cdot \tilde{V}_{1,1,t_0,k}^+ \right) \cdot X_{r-i+1,s-j+1,t_0}^+$$

$$+ \sum_{i,j\in[d],t_0\in[c_0]} \left( \sum_{k\in[2nc_0]} U_{i,j,k,t_1} \cdot \tilde{V}_{1,1,t_0,k}^- \right) \cdot X_{r-i+1,s-j+1,t_0}^-.$$

We remind the reader that $\tilde{S} = S_1 \odot S_2$, with $S_1$ being a $2n$-channel-blocked mask and $S_2$ being a mask that removes filters. Define $L_{i,j,t_0,t_1}^+ = \sum_{k\in[nc]} U_{i,j,k,t_1} \cdot \tilde{V}_{1,1,t_0,k}^+$ and, similarly, $L_{i,j,t_0,t_1}^- = \sum_{k\in[nc]} U_{i,j,k,t_1} \cdot \tilde{V}_{1,1,t_0,k}^-$. Then,

$$\sum_{i,j\in[d],t_0\in[c_0]} \left( \sum_{k\in[nc_0]} U_{i,j,k,t_1} \cdot \tilde{V}_{1,1,t_0,k}^+ \right) \cdot X_{r-i+1,s-j+1,t_0}^+$$

$$+ \sum_{i,j\in[d],t_0\in[c_0]} \left( \sum_{k\in[nc_0]} U_{i,j,k,t_1} \cdot \tilde{V}_{1,1,t_0,k}^- \right) \cdot X_{r-i+1,s-j+1,t_0}^-$$

$$= \sum_{i,j\in[d],t_0\in[c_0]} L_{i,j,t_0,t_1}^+ \cdot X_{r-i+1,s-j+1,t_0}^+ + \sum_{i,j\in[d],t_0\in[c_0]} L_{i,j,t_0,t_1}^- \cdot X_{r-i+1,s-j+1,t_0}^-.$$

We now show that, for each $t_0 \in [c_0]$, $K_{:,:,t_0,:}$ can be $\varepsilon$-approximated by $L_{:,:,t_0,:}^+$ by suitably pruning $\tilde{V}^+$, i.e. by further zeroing entries of $\tilde{S}$, and that such pruning corresponds to solving an instance of MRSS according to Theorem 5. The same reasoning applies to $K^-$ and $L^-$.

For each $t_0 \in [c_0]$, let

$$I_+^{(t_0)} = \left\{ k \in \{ (2t_0 - 2) n + 1, \ldots, (2t_0 - 1) n \} : \tilde{S}_{1,1,t_0,k} = 1 \right\}.$$

Observe that $I_+^{(t_0)}$ consists of the strictly positive entries of $\tilde{V}_{1,1,t_0,:}^+$.[7] Since the entries of $V$ follow a standard normal distribution, each entry is positive with probability $1/2$. By a standard application of Chernoff bounds (Lemma 16 in Appendix A), we then have

$$\Pr\left(\left|I_+^{(t_0)}\right| > \frac{n}{3}\right) \geq 1 - \frac{\varepsilon}{4}, \tag{30}$$

provided that the constant $C$ in the bound on $n$ is sufficiently large.

For each $k \in I_+^{(t_0)}$, up to reshaping the tensor as a one-dimensional vector, $U_{:,:,k,:} \cdot \tilde{V}_{1,1,t_0,k}^+$ is an NSN vector (Definition 4) by Lemma 18 (Appendix B). Thus, for each $t_0 \in [c_0]$ and a sufficiently-large value of $C$, since we have $n \geq C d^{13} c_1^6 \log^3 \frac{d^2 c_1 c_0}{\varepsilon}$, we can apply an amplified version of Theorem 5 (i.e. Corollary 19 in Appendix B with vectors of dimension $d^2 c_1$) to show that, with probability $1 - \frac{\varepsilon}{4c_0}$, for all target filter $K$ such that $\|K_{:,:,t_0,:}\|_1 \leq 1$, there exists a way to zero the entries indexed by $I_+^{(t_0)}$ of $\tilde{S}$ (and thus $\tilde{V}_{1,1,t_0,:}^+$), so that the pruned version of $L_{:,:,t_0,:}^+ = \sum_{k \in [nc_0]} U_{:,:,k,:} \cdot \tilde{V}_{1,1,t_0,k}^+$ approximates $K_{:,:,t_0,:}$. In particular, there exists another binary mask $S_3^+ \in \{0,1\}^{\text{size } \tilde{S}}$ such that $\hat{L}_{:,:,t_0,:}^+ = \sum_{k \in [nc_0]} U_{:,:,k,:} \cdot \hat{V}_{1,1,t_0,k}^+$ approximates $K_{:,:,t_0,:}$, where $\hat{V}^+ = \tilde{V}^+ \odot S_3^+$. An analogous argument carries on for a binary mask $S_3^-$ and $-\hat{L}_{:,:,t_0,:}^-$.[8] More formally, let

$$\mathcal{E}_{t_0,+}^{(\text{kernel})} = \left\{ \forall K : \|K\|_1 \leq 1, \exists S_3^+ \in \{0,1\}^{\text{size } \tilde{S}} \; \middle| \; \left\| \hat{L}_{:,:,t_0,:}^+ - K_{:,:,t_0,:} \right\|_{\max} \leq \frac{\varepsilon}{2d^2 c_1 c_0} \right\},$$

$$\mathcal{E}_{t_0,-}^{(\text{kernel})} = \left\{ \forall K : \|K\|_1 \leq 1, \exists S_3^- \in \{0,1\}^{\text{size } \tilde{S}} \; \middle| \; \left\| \hat{L}_{:,:,t_0,:}^- + K_{:,:,t_0,:} \right\|_{\max} \leq \frac{\varepsilon}{2d^2 c_1 c_0} \right\}, \text{ and}$$

$$\mathcal{E}^{(\text{kernel})} = \left( \bigcap_{t_0 \in [c_0]} \mathcal{E}_{t_0,+}^{(\text{kernel})} \right) \bigcap \left( \bigcap_{t_0 \in [c_0]} \mathcal{E}_{t_0,-}^{(\text{kernel})} \right).$$

Then, by Corollary 19,

$$\Pr\left(\mathcal{E}_{t_0,+}^{(\text{kernel})} \; \middle| \; \left|I_+^{(t_0)}\right| > \frac{n}{3}\right) \geq 1 - \frac{\varepsilon}{4c_0}, \text{ and}$$

$$\Pr\left(\mathcal{E}_{t_0,-}^{(\text{kernel})} \; \middle| \; \left|I_-^{(t_0)}\right| > \frac{n}{3}\right) \geq 1 - \frac{\varepsilon}{4c_0}.$$

By the union bound, we have the following:

$$\Pr\left(\mathcal{E}^{(\text{kernel})} \; \middle| \; \left|I_+^{(t_0)}\right|, \left|I_-^{(t_0)}\right| > \frac{n}{3}\right)$$

$$= 1 - \Pr\left( \left( \bigcup_{t_0 \in [c_0]} \overline{\mathcal{E}}_{t_0,+}^{(\text{kernel})} \right) \bigcup \left( \bigcup_{t_0 \in [c_0]} \overline{\mathcal{E}}_{t_0,-}^{(\text{kernel})} \right) \; \middle| \; \left|I_+^{(t_0)}\right|, \left|I_-^{(t_0)}\right| > \frac{n}{3} \right)$$

$$\geq 1 - \sum_{t_0 \in [c_0]} \left[ \Pr\left( \overline{\mathcal{E}}_{t_0,+}^{(\text{kernel})} \; \middle| \; \left|I_+^{(t_0)}\right|, \left|I_-^{(t_0)}\right| > \frac{n}{3} \right) + \Pr\left( \overline{\mathcal{E}}_{t_0,-}^{(\text{kernel})} \; \middle| \; \left|I_+^{(t_0)}\right|, \left|I_-^{(t_0)}\right| > \frac{n}{3} \right) \right]$$

$$\geq 1 - 2 \sum_{t_0 \in [c_0]} \frac{\varepsilon}{4c_0}$$

$$\geq 1 - \frac{\varepsilon}{2}.$$

Since $\Pr\left(\left|I_+^{(t_0)}\right|, \left|I_-^{(t_0)}\right| > \frac{n}{3}\right) \geq 1 - \frac{\varepsilon}{2}$, then we can remove the conditional event obtaining

$$\Pr\left(\mathcal{E}^{(\text{kernel})}\right) \geq \Pr\left(\mathcal{E}^{(\text{kernel})} \; \middle| \; \left|I_+^{(t_0)}\right|, \left|I_-^{(t_0)}\right| > \frac{n}{3}\right) \Pr\left(\left|I_+^{(t_0)}\right|, \left|I_-^{(t_0)}\right| > \frac{n}{3}\right)$$

---

[7]Notice that excluding zero entries implies conditioning on the event that the entry is not zero. However, such an event has zero probability and, thus, does not impact the analysis.

[8]The negative sign in front of $\hat{L}_{:,:,t_0,:}^-$ does not affect the random subset sum result as each entry is independently negative or positive with the same probability.

$$\geq \left(1 - \frac{\varepsilon}{2}\right)^2$$
$$\geq 1 - \varepsilon. \tag{31}$$

To rewrite the latter in terms of the filter $K$ and a mask $\hat{S}$, we notice that pruning $L^+_{:,:,t_0,:}$ and $L^-_{:,:,t_0,:}$ separately, with two binary masks, is equivalent to say that there exists a single binary mask $S_3 \in \{0,1\}^{\text{size } \tilde{S}}$ such that, $\hat{L}_{:,:,t_0,:}$ can be written as $\hat{L}_{:,:,t_0,:} = \sum_{k \in [nc_0]} U_{:,:,k,:} \cdot \hat{V}_{1,1,t_0,k}$, where $\hat{V} = \tilde{V} \odot S_3$. Ineq. 31 implies that, with probability $1 - \varepsilon$, for all target filters $K$ with $\|K\|_1 \leq 1$ such $S_3$ exists and hence,

$$\left\|K - \hat{L}^+\right\|_{\max} + \left\|K + \hat{L}^-\right\|_{\max} \leq \frac{\varepsilon}{d^2 c_1 c_0}. \tag{32}$$

Let $\hat{S} = \tilde{S} \odot S_3$: then $\hat{S} = S_1 \odot S_2 \odot S_3$ is the combination of a $2n$-channel-blocked mask $S = S_1$ and a mask $S_2 \odot S_3$ that only removes filters.

Also, whenever such binary mask $\hat{S}$ exists, we have that

$$\sup_{X : \|X\|_{\max} \leq M} \left\|K * X - N_0^{(\hat{S})}(X)\right\|_{\max}$$
$$= \sup_{X : \|X\|_{\max} \leq M} \left\|K * X - U * \phi\left((V \odot \hat{S}) * X\right)\right\|_{\max}$$
$$= \sup_{X : \|X\|_{\max} \leq M} \left\|K * X - U * \phi\left((V \odot \tilde{S} \odot S_3) * X\right)\right\|_{\max}$$
$$= \sup_{X : \|X\|_{\max} \leq M} \left\|K * (X^+ - X^-) - U * \left(\left(\hat{V}^+ * X^+\right) + \left(\hat{V}^- * X^-\right)\right)\right\|_{\max},$$

where the latter holds by Lemma 11.[9] Then, by the distributive property of the convolution and the triangle inequality,

$$\sup_{X : \|X\|_{\max} \leq M} \left\|K * (X^+ - X^-) - U * \left(\left(\hat{V}^+ * X^+\right) + \left(\hat{V}^- * X^-\right)\right)\right\|_{\max}$$
$$= \sup_{X : \|X\|_{\max} \leq M} \left\|K * X^+ - U * \left(\hat{V}^+ * X^+\right) - K * X^- - U * \left(\hat{V}^- * X^-\right)\right\|_{\max}$$
$$\leq \sup_{X : \|X\|_{\max} \leq M} \left\|K * X^+ - U * \left(\hat{V}^+ * X^+\right)\right\|_{\max}$$
$$+ \sup_{X : \|X\|_{\max} \leq M} \left\|K * X^- + U * \left(\hat{V}^- * X^-\right)\right\|_{\max}.$$

One can now apply the Tensor Convolution Inequality (Lemma 20) and obtain

$$\sup_{X : \|X\|_{\max} \leq M} \left\|K * X^+ - U * \left(\hat{V}^+ * X^+\right)\right\|_{\max}$$
$$+ \sup_{X : \|X\|_{\max} \leq M} \left\|K * X^- + U * \left(\hat{V}^- * X^-\right)\right\|_{\max}$$
$$\leq \sup_{X : \|X\|_{\max} \leq M} \left\|X^+\right\|_{\max} \cdot \left\|K - U * \hat{V}^+\right\|_1$$
$$+ \sup_{X : \|X\|_{\max} \leq M} \left\|X^-\right\|_{\max} \cdot \left\|K + U * \hat{V}^-\right\|_1$$
$$= M \cdot \left\|K - U * \hat{V}^+\right\|_1 + M \cdot \left\|K + U * \hat{V}^-\right\|_1.$$

Now, observing that the number of entries of the two tensors in the expression above is $d^2 c_1 c_0$, and using Ineq. 32, we get that

$$M \cdot \left\|K - U * \hat{V}^+\right\|_1 + M \cdot \left\|K - U * \hat{V}^-\right\|_1$$

---

[9]The presence of $S_3$ does not influence the proof of Lemma 11.

$$\leq d^2 c_1 c_0 \left( \left\| K - U * \hat{V}^+ \right\|_{\max} + \left\| K - U * \hat{V}^- \right\|_{\max} \right)$$

$$\leq d^2 c_1 c_0 M \frac{\varepsilon}{d^2 c_1 c_0}$$

$$= \varepsilon M.$$

proving the thesis. $\qquad\square$

*Proof of Theorem 2.* To bound the error propagation across layers, we define the layers' outputs

$$X^{(0)} = X,$$
$$X^{(i)} = \phi \left( K^{(i)} * X^{(i-1)} \right) \quad \text{for } 1 \leq i \leq \ell. \tag{33}$$

Notice that $X^{(\ell)}$ is the output of the target function, i.e., $f(X) = X^{(\ell)}$.

For brevity's sake, given masks $S^{(2i-1)} \in \{0,1\}^{\text{size}(L^{(2i-1)})}$ for each $i = 1, \dots, \ell$, let us denote

$$\tilde{L}^{(2i-1)} = L^{(2i-1)} \odot S^{(2i-1)}. \tag{34}$$

We now show that the random network approximates the target one after being pruned only at layers $L^{(2i-1)}$ for all $i = 1, \dots, \ell$, by masks $S^{(2i-1)}$ which are the composition of a $2n_{2i-1}$-channel-blocked mask and a mask that only removes filters. The key idea is to iteratively applying Lemma 12. Since the ReLU function is 1-Lipschitz, for all $i$ it holds

$$\left\| \phi \left( K^{(i)} * X^{(i-1)} \right) - \phi \left( L^{(2i)} * \phi \left( \tilde{L}^{(2i-1)} * X^{(i-1)} \right) \right) \right\|_{\max}$$
$$\leq \left\| K^{(i)} * X^{(i-1)} - L^{(2i)} * \phi \left( \tilde{L}^{(2i-1)} * X^{(i-1)} \right) \right\|_{\max}. \tag{35}$$

The key step of the proof is that, for each layer $i$, since $n_i \geq C d^{13} c_i^6 \log^3 \frac{d^2 c_i c_{i-1} \ell}{\varepsilon}$ for a suitable constant $C$, we can apply Lemma 12 to get that, with probability at least $1 - \frac{\varepsilon}{2\ell}$, for all $i$ and for all choices of target filters $K^{(i)} \in \mathbb{R}^{d_i \times d_i \times c_{i-1} \times c_i}$ it holds

$$\left\| K^{(i)} * X^{(i-1)} - L^{(2i)} * \phi \left( \tilde{L}^{(2i-1)} * X^{(i-1)} \right) \right\|_{\max} < \frac{\varepsilon}{2\ell} \cdot \left\| X^{(i-1)} \right\|_{\max}. \tag{36}$$

Hence, combining Eq. 35 and Eq. 36 we get that, with probability at least $1 - \frac{\varepsilon}{2\ell}$, for all $i$ and all choices of target filters $K^{(i)} \in \mathbb{R}^{d_i \times d_i \times c_{i-1} \times c_i}$ with $\left\| K^{(i)} \right\|_1 \leq 1$,

$$\left\| \phi \left( K^{(i)} * X^{(i-1)} \right) - \phi \left( L^{(2i)} * \phi \left( \tilde{L}^{(2i-1)} * X^{(i-1)} \right) \right) \right\|_{\max}$$
$$< \frac{\varepsilon}{2\ell} \cdot \left\| X^{(i-1)} \right\|_{\max}. \tag{37}$$

By a union bound, with probability at least $1 - \varepsilon$, we get that Eq. 37 holds for all layers $i$ and all choices of target filters $K^{(i)} \in \mathbb{R}^{d_i \times d_i \times c_{i-1} \times c_i}$.

Analogously, we can define the pruned layers' outputs

$$\tilde{X}^{(0)} = X,$$
$$\tilde{X}^{(i)} = \phi \left( L^{(2i)} * \phi \left( \tilde{L}^{(2i-1)} * \tilde{X}^{(i-1)} \right) \right) \quad \text{for } 1 \leq i \leq \ell. \tag{38}$$

Notice that $\tilde{X}^{(\ell)}$ is the output of the pruned network, i.e. $N_0^{\left( S^{(1)}, \dots, S^{(2\ell)} \right)} (X) = \tilde{X}^{(\ell)}$.

By the same reasoning employed to derive Eq. 36 and Eq. 37 we have that, with probability $1 - \varepsilon$, for all layers $i$ and all choices of target filters $K^{(i)} \in \mathbb{R}^{d_i \times d_i \times c_{i-1} \times c_i}$, the output of all pruned layers satisfies

$$\left\| \phi \left( K^{(i)} * \tilde{X}^{(i-1)} \right) - \phi \left( L^{(2i)} * \phi \left( \tilde{L}^{(2i-1)} * \tilde{X}^{(i-1)} \right) \right) \right\|_{\max}$$
$$< \frac{\varepsilon}{2\ell} \cdot \left\| \tilde{X}^{(i-1)} \right\|_{\max}. \tag{39}$$

Moreover, for each $1 \leq i \leq \ell - 1$, by the triangle inequality and by Eq. 39,

$$\left\| \tilde{X}^{(i)} \right\|_{\max} = \left\| \tilde{X}^{(i)} - \phi\left(K^{(i)} * \tilde{X}^{(i-1)}\right) + \phi\left(K^{(i)} * \tilde{X}^{(i-1)}\right) \right\|_{\max}$$

$$\leq \left\| \tilde{X}^{(i)} - \phi\left(K^{(i)} * \tilde{X}^{(i-1)}\right) \right\|_{\max} + \left\| \phi\left(K^{(i)} * \tilde{X}^{(i-1)}\right) \right\|_{\max}$$

$$\leq \frac{\varepsilon}{2\ell} \cdot \left\| \tilde{X}^{(i-1)} \right\|_{\max} + \left\| \phi\left(K^{(i)} * \tilde{X}^{(i-i)}\right) \right\|_{\max}.$$

By the Lipschitz property of $\phi$ and Lemma 20

$$\frac{\varepsilon}{2\ell} \cdot \left\| \tilde{X}^{(i-1)} \right\|_{\max} + \left\| \phi\left(K^{(i)} * \tilde{X}^{(i-i)}\right) \right\|_{\max}$$

$$\leq \frac{\varepsilon}{2\ell} \cdot \left\| \tilde{X}^{(i-1)} \right\|_{\max} + \left\| K^{(i)} * \tilde{X}^{(i-i)} \right\|_{\max}$$

$$\leq \frac{\varepsilon}{2\ell} \cdot \left\| \tilde{X}^{(i-1)} \right\|_{\max} + \left\| K^{(i)} \right\|_1 \left\| \tilde{X}^{(i-i)} \right\|_{\max}$$

$$= \left\| \tilde{X}^{(i-1)} \right\|_{\max} \left(1 + \frac{\varepsilon}{2\ell}\right).$$

By unrolling the recurrence, we get that, with probability $1 - \varepsilon$, for all choices of target filters $K^{(i)} \in \mathbb{R}^{d_i \times d_i \times c_{i-1} \times c_i}$,

$$\left\| \tilde{X}^{(i)} \right\|_{\max} \leq \left\| \tilde{X}^{(0)} \right\|_{\max} \left(1 + \frac{\varepsilon}{2\ell}\right)^i. \tag{40}$$

Thus, combining Eq. 39 and Eq. 40, with probability $1 - \varepsilon$ we get that, for each $i \in [\ell]$ and for all choices of target filters $K^{(i)} \in \mathbb{R}^{d_i \times d_i \times c_{i-1} \times c_i}$,

$$\left\| K^{(i)} * \tilde{X}^{(i-1)} - L^{(2i)} * \phi\left(\tilde{L}^{(2i-1)} * \tilde{X}^{(i-1)}\right) \right\|_{\max}$$

$$< \frac{\varepsilon}{2\ell} \cdot \left(1 + \frac{\varepsilon}{2\ell}\right)^{i-1} \left\| \tilde{X}^{(0)} \right\|_{\max}. \tag{41}$$

We then see that with probability $1 - \varepsilon$, for $1 \leq i \leq \ell$ and for all choices of target filters $K^{(i)} \in \mathbb{R}^{d_i \times d_i \times c_{i-1} \times c_i}$, by Eq. 33 and Eq. 38, and by the triangle inequality,

$$\left\| X^{(\ell)} - \tilde{X}^{(\ell)} \right\|_{\max}$$

$$= \left\| \phi\left(K^{(\ell)} * X^{(\ell-1)}\right) - \phi\left(L^{(2\ell)} * \phi\left(\tilde{L}^{(2\ell-1)} * \tilde{X}^{(\ell-1)}\right)\right) \right\|_{\max}$$

$$\leq \left\| \phi\left(K^{(\ell)} * X^{(\ell-1)}\right) - \phi\left(K^{(\ell)} * \tilde{X}^{(\ell-1)}\right) \right\|_{\max}$$

$$+ \left\| \phi\left(K^{(\ell)} * \tilde{X}^{(\ell-1)}\right) - \phi\left(L^{(2\ell)} * \phi\left(\tilde{L}^{(2\ell-1)} * \tilde{X}^{(\ell-1)}\right)\right) \right\|_{\max}.$$

Again by the 1-Lipschitz property of the ReLU activation function, and by the distributive property of the convolution operation,

$$\left\| \phi\left(K^{(\ell)} * X^{(\ell-1)}\right) - \phi\left(K^{(\ell)} * \tilde{X}^{(\ell-1)}\right) \right\|_{\max}$$

$$+ \left\| \phi\left(K^{(\ell)} * \tilde{X}^{(\ell-1)}\right) - \phi\left(L^{(2\ell)} * \phi\left(\tilde{L}^{(2\ell-1)} * \tilde{X}^{(\ell-1)}\right)\right) \right\|_{\max}$$

$$\leq \left\| K^{(\ell)} * X^{(\ell-1)} - K^{(\ell)} * \tilde{X}^{(\ell-1)} \right\|_{\max}$$

$$+ \left\| K^{(\ell)} * \tilde{X}^{(\ell-1)} - L^{(2\ell)} * \phi\left(\tilde{L}^{(2\ell-1)} * \tilde{X}^{(\ell-1)}\right) \right\|_{\max}$$

$$= \left\| K^{(\ell)} * \left(X^{(\ell-1)} - \tilde{X}^{(\ell-1)}\right) \right\|_{\max}$$

$$+ \left\| K^{(\ell)} * \tilde{X}^{(\ell-1)} - L^{(2\ell)} * \phi\left(\tilde{L}^{(2\ell-1)} * \tilde{X}^{(\ell-1)}\right) \right\|_{\max}.$$

Lemma 20 and the hypothesis $\left\| K^{(\ell)} \right\|_1 \leq 1$ imply that

$$\left\| K^{(\ell)} * \left(X^{(\ell-1)} - \tilde{X}^{(\ell-1)}\right) \right\|_{\max}$$

$$+ \left\| K^{(\ell)} * \tilde{X}^{(\ell-1)} - L^{(2\ell)} * \phi\left(\tilde{L}^{(2\ell-1)} * \tilde{X}^{(\ell-1)}\right) \right\|_{\max}$$

$$\leq \left\| K^{(\ell)} \right\|_1 \cdot \left\| \left(X^{(\ell-1)} - \tilde{X}^{(\ell-1)}\right) \right\|_{\max}$$

$$+ \left\| K^{(\ell)} * \tilde{X}^{(\ell-1)} - L^{(2\ell)} * \phi\left(\tilde{L}^{(2\ell-1)} * \tilde{X}^{(\ell-1)}\right) \right\|_{\max}$$

$$\leq \left\| \left(X^{(\ell-1)} - \tilde{X}^{(\ell-1)}\right) \right\|_{\max}$$

$$+ \left\| K^{(\ell)} * \tilde{X}^{(\ell-1)} - L^{(2\ell)} * \phi\left(\tilde{L}^{(2\ell-1)} * \tilde{X}^{(\ell-1)}\right) \right\|_{\max}.$$

Now, we first apply Eq. 41 and then we unroll the recurrence for all layers (as, with probability $1 - \varepsilon$, Eq. 41 holds for all layers and for all choices of target filters $K^{(i)} \in \mathbb{R}^{d_i \times d_i \times c_{i-1} \times c_i}$), obtaining

$$\left\| \left(X^{(\ell-1)} - \tilde{X}^{(\ell-1)}\right) \right\|_{\max}$$

$$+ \left\| K^{(\ell)} * \tilde{X}^{(\ell-1)} - L^{(2\ell)} * \phi\left(\tilde{L}^{(2\ell-1)} * \tilde{X}^{(\ell-1)}\right) \right\|_{\max}$$

$$\leq \left\| X^{(\ell-1)} - \tilde{X}^{(\ell-1)} \right\|_{\max} + \frac{\varepsilon}{2\ell} \cdot \left(1 + \frac{\varepsilon}{2\ell}\right)^{\ell-1}$$

$$\leq \sum_{j=1}^{\ell} \frac{\varepsilon}{2\ell} \cdot \left(1 + \frac{\varepsilon}{2\ell}\right)^{j-1}.$$

By summing the geometric series and observing that $\varepsilon < 1$, we conclude that

$$\sum_{j=1}^{\ell} \frac{\varepsilon}{2\ell} \cdot \left(1 + \frac{\varepsilon}{2\ell}\right)^{j-1} = \left(1 + \frac{\varepsilon}{2\ell}\right)^{\ell} - 1$$

$$\leq e^{\frac{\varepsilon}{2}} - 1$$

$$\leq \varepsilon.$$

Hence,

$$\left\| X^{(\ell)} - \tilde{X}^{(\ell)} \right\|_{\max} \leq \varepsilon,$$

yielding the thesis. $\qquad\square$

