# OpenReview forum: "Polynomially Over-Parameterized Convolutional Neural Networks Contain Structured Strong Winning Lottery Tickets"
_NeurIPS.cc/2023/Conference — NeurIPS 2023 poster_

### Official Review · Reviewer_Up5n · 2023-06-17

**Soundness:** 3 good
**Presentation:** 3 good
**Contribution:** 3 good
**Rating:** 6
**Confidence:** 3

**Summary:**

The central theme of this paper is the Strong Lottery Ticket Hypothesis (SLTH), which conjectures that « randomly initialised networks contain sparse subnetworks […] that perform well without any training » (lines 26-30). Pruning large networks in order to obtain those « strong lottery tickets » would be a way to limit the computational costs associated to those over-parameterised networks and facilitate their deployment in real-world applications. The SLTH has already been theoretically verified for various classes of networks, but never while restraining the way the pruning must be done in order to obtain those « strong lottery tickets ». The authors aim to theoretically verify that « strong lottery tickets » can be obtained via structural pruning of a certain class of convolutional neural networks. The way they do it is by proving that « networks in a wide class of CNNs are likely to contain structured subnetworks that approximate any sufficiently smaller CNN in the class » (lines 77-79), based on the assumption that their exists a «  sufficiently smaller CNN in the class » that performs well.

**Strengths:**

1. The manipulated concepts, such as the LTH, the SLTH, or the RSSP are well-explained.

2. A clear understanding of the many proofs can be obtained by reading Section 4.

**Weaknesses:**

_Major_

1. The proof ideas are not that novel. Many cited works from the Related Work section use the same technique in order to obtain similar results, and Theorems 3 and 5 are the only contributions of the paper (lines 81-85; see Question 3).

2. The goal is to demonstrate that winning tickets from the SLTH can be obtained by removing channels from a deep and wide CNN. The underlying assumption here being that there exists a CNN having the same restrictions as discussed in the article AND that is significantly smaller than an over-parameterised one is in fact capable of obtaining good results on given tasks. Considering those limitations (no classification head, that is, no fully-connected layer; no pooling layer; no batch norm layer; ReLU activations only; etc.), such an assumption is not trivial and questions the whole process.

3. The justification for the approach is that for over-parameterised networks, « [their] associated computational cost limits both the progress of [the neural networks in general] and their deployment in real-world applications. » (lines 19-20). Therefore, the justification is on a practical level. But considering the size of the kernels needed for the network to be pruned (with $n_i \geq Cd^{12}c^6_i ...$, line 139), from the practical point of view, it is virtually impossible to encode such a network even before considering pruning it (which would then be a fastidious process and would imperatively be done in an efficient manner, which is not discussed in the article).

4. The end-goal is to obtain a relatively small network, but taking $n_i \geq Cd^{12}c^6_i ...$ (line 139), and plugging it into $m_i = \sqrt{n_i / (C₁ log \frac{1}{e})}$ (line 150) leads to $m_i \geq d^6c_i^3...$, which do not seem to produce a network that would be smaller.



_Typos / Minor_

1. The way the citations are made is inconsistent throughout the paper; their are both appropriate (« The constructions proposed by Burkholz [2022a,b] », line 290) and inappropriate (« both in the past Reed [1993] and in the present [...]. », line 21) inline text citation format.

2. Line 58 : or block sparsity Siswanto [2021] (Figure 1$\underline{c}$).

3. Some statements are not supported; for example : « Much of the success of deep learning techniques relies on extreme over-parameterisation. » (line 17).

**Questions:**

1. Exactly which lemmas are original and which have been taken from the literature?

2. Is there a reason why you specifically chose having filters (or channels) being removed as structured pruning technique?

3. The third contribution states that « Additionally, our pruning scheme focuses on filter pruning, which, like neuron pruning, allows for a direct reduction of the size of the original CNN » (lines 86-87) (which is not a contribution, but an argument assessing the relevance of the work). It is also stated that « through unstructured pruning [...] one can usually reach 95% sparsity without accuracy loss » (lines 54-55). One could argue that by reaching 95% sparsity, the size of the original network is likely to diminish too. From that perspective, why would channel-pruning still be more desirable?

**Limitations:**

Limitations have been discussed properly, apart from what has been raised in the Weaknesses part of the review.

---

> ### Author Rebuttal · Authors · 2023-08-09
>
> We are grateful to the reviewer for their time and valuable comments. We hope to clarify some issues brought up by the reviewer.
>
> Weaknesses (major)
> 1. While our proof is based on a well-known probabilistic technique (the second moment method), our analysis is entirely novel, except for the Young’s Convolution Inequality (underlying Lemma 20) and the technical tools in SM A.1 (Lemmas 14 to 17). All other lemmas and corollaries are new, including those in the main paper. Most notably, our work explores the MRSSP in the presence of dependencies, which is of interest on its own, as appreciated by other reviewers. Despite being similar in spirit to previous works [Borst et al. 2022; Becchetti et al. 2022], the need to overcome stochastic dependencies required new proof strategies in all lemmas related to the SSP. Moreover, we remark that the average behaviour of the multi-dimensional SSP is an active field of research and that attempts to tackle it with more sophisticated probabilistic tools have, so far, not been fruitful (see, e.g., section 3.1 of Becchetti et al. [2022]). As for the pruning section, despite apparent similarities with previous works (e.g., Pensia et al. [2020], da Cunha et al. [2022], and Burkholz [2022]) in applying subset sum results to convolutions, the change from unstructured to structured pruning requires a different formalisation of the problem. Those arguments are closely tied to the structures obtained when pruning the network and we had to devise a new one to obtain a set up that allowed for selecting and summing subsets of entire filters.
> 2. We highlight that convolutional layers are quite expressive and generalise many other transformations, such as dense layers, batch normalisation and average pooling. Our restriction to ReLU activations is in line with most works in the field and, as discussed with Reviewer ryBt, there are reasons to believe it can be relaxed.
> We also believe that an alternative (but equivalent) framing of our results may make the relevance of our contribution more clear to the reviewer. For **any** given random CNN, we wish to know how well it can perform after pruning (without any training). Our results imply that, under suitable hypothesis, it can perform almost as well (if not better) as **the best** CNN among all those that are sufficiently smaller than the random CNN and whose weights can be tuned. In particular, such class includes traditionally trained networks.
> 3. (and 4) Please refer to points 1 and 2 of the general answer.
>
> Weaknesses (minor)
> We thank the reviewer for the notes about minor issues. We have addressed all of them.
>
> Questions
> 1. Please refer to the answer to weakness number 1.
> 2. (and 3) We attempted to convey the reasoning behind it in our introduction (mainly in the paragraph starting at line 51), but we acknowledge that this aspect can indeed be counter-intuitive. Concentrating on dense layers for the sake of simplicity, the key point is that making an $n \times n$ matrix 95% sparse does not translate directly into a speed-up of the associated matrix multiplication. For example, if one naively performs the multiplication directly, there are no computational gains whatsoever (the BLAS pipeline is oblivious to the sparsity of the matrix). Of course there are nuances to the matter, as we discuss in the introduction. On the other hand, if the sparsity in the matrix is conveniently structured, even if the degree of sparsity is much lower, the sparsity directly translates into computational performance, potentially offsetting the difference in sparsity. E.g., if 50% of the rows/columns are filled with zeros, rather than storing those rows/columns, we can simply drop them to obtain a (still dense) smaller matrix. Doing so requires no memory overhead, directly translating into a 50% reduction in the memory footprint, while also speeding up the associated matrix multiplication regardless of software/hardware intricacies.
>
> We hope that we have responded satisfactorily to the reviewer's comments and that our response will lead them to appreciate our work more. We look forward to answering any questions the reviewer may still have.

---

> > ### Comment · Reviewer_Up5n · 2023-08-14
> >
> > Thank you for the insightful comments. I might originally have viewed the submission from the wrong angle, being too concerned by the practical issues of the presented bounds, maybe because of the justifications raised by the authors for their work. The "real-world application" therefore is not an argument to support the obtained results, but to support research in this particular avenue. It is true that convolutional layers indeed can represent other types of layers (e.g. dense layer, average pooling). Also, it is now clearer to me what results are original contributions.
> >
> > I'm still having a concern with the following:
> >
> > Weaknesses (major)
> >
> > 2 - As for the use of ReLU activation function, the authors mentioned in their rebuttal that "[..] as discussed with Reviewer ryBt, there are reasons to believe it can be relaxed." The discussion with reviewer ryBt highlights that the authors $\underline{\textup{think}}$ there are reasons to believe it can be relaxed, not what those reasons are.
> >
> > Question
> >
> > When it comes to a dense layer represented by a convolutional layer, removing a filter in the latter is similar to removing a neuron in the former. Is there any work concerning this type of structured pruning for fully-connected layers/networks? If so, how do these results align with the ones from the manuscript?
> >
> > Nevertheless, I'm satisfied with the answers given by the authors to my concerns and the discussion they had/are having with the other reviewers. I'm modifying my score accordingly.

---

> > > ### Author Response · Authors · 2023-08-16
> > >
> > > We are grateful to the reviewer for his time and valuable questions.
> > >
> > > Regarding the use of other activation functions, we apologize for our loose language. We meant that even though the arguments present in [Burkholz 2022] do not translate to our setting, it seemed to us that a deeper rework of those ideas could suffice. More precisely, the proof by Burkholz revolves around the identity $x = \phi(x) - \phi(-x)$ and the fact that the SLTH arguments used for the dense case are robust to small perturbations of this identity. In our case, one can see a loose analogy between the identity above and Lemma 11, which we also believe to be robust to small perturbation of the activation function. However, this is just a first impression and we have not formally explored this possibility.
> > >
> > > Regarding the reviewer's question, to the best of our knowledge, only the work by Malach et al. [2020] addresses neuron pruning of dense networks in the context of SLTH. We have discussed this with reviewer TQCH and report the main points here for your convenience.
> > >
> > > Under reasonable assumptions, Malach et al. proved that pruning neurons in a fully connected feed-forward neural network of two layers (one hidden layer) is equivalent to training the random feature model. Important constraints on the expressiveness of the latter model are known: Yehudai and Shamir [2019] proved that approximating even a single ReLU neuron under the random features model requires either exponentially large weights or an exponentially wide hidden layer.
> > > Our work investigates alternative forms of structured sparsity instead of pure neuron pruning.
> > > Namely, we use a type of block-sparsity to induce the structure described in our definition  of *channel-blocked mask* (definition 2) before performing the filter removals (the CNN generalization of neuron removal).
> > >
> > > For neuron/filter pruning in general, several papers have investigated the relationship between compression rate and accuracy loss before, during, or after training. A paper that provides state-of-the-art results as well as many references to recent related literature is [Tukan et al. 2022]. To the best of our knowledge, there is no generalization of training-by-pruning methods such as edge pop-up [Ramanujan et al. 2020] to the structured case.
> > >
> > > We hope we have provided a satisfactory answer to the reviewer's remaining points, and we are happy to answer any further questions the reviewer may have.
> > >
> > > Burkholz [2022]: Most activation functions can win the lottery without excessive depth, Rebekka Burkholz, NeurIPS 2022.
> > >
> > > Malach et al. [2020]: Proving the Lottery Ticket Hypothesis: Pruning is All You Need, Eran Malach, Gilad Yehudai, Shai Shalev-shwartz, and Ohad Shamir, ICML 2020.
> > >
> > > Yehudai and Shamir [2019]: On the Power and Limitations of Random Features for Understanding Neural Networks, Gilad Yehudai and Ohad Shamir, NeurIPS 2019.
> > >
> > > Tukan et al. [2022]: Pruning Neural Networks via Coresets and Convex Geometry: Towards No Assumptions, Murad Tukan, Loay Mualem, and Alaa Maalouf, NeurIPS 2022.
> > >
> > > Ramanujan et al. [2020]: What’s Hidden in a Randomly Weighted Neural Network?, Vivek Ramanujan, Mitchell Wortsman, Aniruddha Kembhavi, Ali Farhadi, and Mohammad Rastegari, CVPR 2020.

---

> > > > ### Comment · Reviewer_Up5n · 2023-08-18
> > > >
> > > > I thank the authors for the clarifications and the interesting comparisons. This answers all of my questions.

---

### Official Review · Reviewer_Ewpx · 2023-06-29

**Soundness:** 3 good
**Presentation:** 3 good
**Contribution:** 3 good
**Rating:** 5
**Confidence:** 1

**Summary:**

This paper tries to discuss the Strong Lottery Ticket Hypothesis (SLTH) in the context of structured pruning in convolutional neural networks (CNNs). The SLTH suggests that randomly-initialized neural networks possess subnetworks that can perform well without any training. While unstructured pruning has been extensively studied, the paper explores the less explored area of structured pruning, which can offer significant computational and memory efficiency gains. The authors overcome the limitations of existing mathematical tools by leveraging recent advancements in the multidimensional generalization of the Random Subset-Sum Problem. By introducing a variant that accounts for stochastic dependencies in structured pruning, the paper proves the existence of structured subnetworks in random CNNs that can approximate smaller networks. This theoretical contribution extends the understanding of the role of overparameterization in deep learning and opens new avenues for further research on the SLTH for structured pruning.

**Strengths:**

- The paper addresses an important research gap by focusing on structured pruning in the context of the Strong Lottery Ticket Hypothesis. This expands the understanding of lottery ticket mechanisms beyond unstructured pruning.
- The authors make use of recent advancements in the multidimensional generalization of the Random Subset-Sum Problem, demonstrating their ability to leverage and adapt existing mathematical tools to overcome limitations in formal analyses of the SLTH.

**Weaknesses:**

The SLTH theory is important in both the pruning and neural network architecture search domains. However, I'm not familiar with the specific analysis methods proposed in this paper, so I can't provide professional advice.

- Line 118, "Preliminaries and contribution" -> "Preliminaries"?
- In this paper, if the convolution kernel $K$ relies on the normality assumption, as mentioned in Line 131, I suggest the author can cite the Convolution Weight Distribution Assumption discussed in [1].
- If possible, I recommend that the author use more intuitive language when writing Section 4 and the introduction. This would help researchers in the pruning and neural network architecture search fields to better understand the contributions of this paper.

[1] Huang Z, Shao W, Wang X, et al. Rethinking the pruning criteria for convolutional neural network[J]. Advances in Neural Information Processing Systems (NeurIPS), 2021, 34: 16305-16318.

**Questions:**

See weakness.

**Limitations:**

The limitations of the proposed analysis have been discussed in this paper.

---

> ### Author Rebuttal · Authors · 2023-08-09
>
> We thank the reviewer for their time and the comments.
>
> Weaknesses
> 1. We can change the section’s title but we remark that in Section 3 we state our contribution more formally and, to do so, we need to introduce some preliminaries.
> 2. We thank the reviewer for the reference. We will add it to the text.
> 3. We will try to improve it.

---

> > ### Comment · Reviewer_Ewpx · 2023-08-17
> > **Re: Rebuttal by Authors**
> >
> > Thank you for your reply and I have also read the comments of other reviewers. I feel like maintaining my score.

---

### Official Review · Reviewer_TQCH · 2023-07-03

**Soundness:** 2 fair
**Presentation:** 2 fair
**Contribution:** 3 good
**Rating:** 5
**Confidence:** 3

**Summary:**

The paper investigates a structured version of strong lottery ticket hypothesis (SLTH) for CNNs, which is important for its computational efficiency against the standard unstructured counterpart. For this purpose, they prove a variant of subset-sum lemma for their situation. Using their subset-sum lemma and splitting lemma for ReLU activations, SLTH for CNNs is proven by combining block-pattern pruning and neuron pruning under the assumption that the network size is polynomially larger than a target network.

**Strengths:**

1. SLTH by structured pruning is an important topic in terms of applicability and computational efficiency of SLTH. The authors are the first to prove the structured version of SLTH for CNNs.

2. They extend the subset-sum lemma, which is a key lemma for SLTH with logarithmic over-parameterization as shown in Pensia et al. 2020, for their purpose.

**Weaknesses:**

Presentation:

1. Although the overview of their proof is straightforward, the presented proof does not provide any intuition for the evaluation of the required network size.

2. No experiments are provided. Since the structured version of SLTH itself has not yet observed empirically even for MLPs or CNNs, some experiments would be helpful for supporting their claim.

3. They claimed that `To the best of our knowledge, this is the first result around the SLTH for structured pruning of neural networks of any kind` in Introduction, but the previous work by Malach et al. 2020 already provided results on the equivalence between SLTH by neuron pruning and random feature NNs in the 2-layer case.

Results:

4. Even though they use a variant of subset-sum lemma following the approach by Pensia et al. 2020 for logarithmic over-parameterization, the required width in this paper is still polynomially larger than the target one. In this point, I wonder that, If one allows such polynomially large width, the structured version of SLTH might be easily proven by simply extending the previous discussion in Malach et al. or Pensia et al., at least for MLPs. Also I wonder that whether their claim is still valid in practice due to such polynomial over-parameterization. Nevertheless, the first attempt to prove the structured version of SLTH is worthful in itself. I hope the authors would emphasise which technique in their proof is important for either structured pruning or handling CNNs.

**Questions:**

1. Can the proof also work for the easier case of MLPs instead of CNNs? Is the variant of subset-sum lemma in this paper also needed even for MLPs? If so, providing some intuition for MLP case may be helpful to understand the structured SLTH. If not, discussing why your version of subset-sum lemma is needed for CNNs would be helpful for readers.

2. Is the structured version of SLTH with polynomial over-parameterization really practical? I wonder if we can find such a structured subnetwork in randomly initialized CNNs (or MLPs) in a similar way to Ramanujan et al. 2020, where the unstructured version of SLTH is actually observed.

**Limitations:**

Yes.

---

> ### Author Rebuttal · Authors · 2023-08-09
>
> We are thankful to the reviewer for their time and comments. We address all main concerns.
>
> Weaknesses
> 1. We apologise, but we are not sure we understood the issue brought by the reviewer. The polynomial overparameterization comes from recent results on the multidimensional random subset sum problem for independent input random vectors ([Borst et al. 2022; Becchetti et al. 2022]) which we extended to the case of dependent inputs (Corollary 19 in SM A) and used in Lemma 12. We remark that Corollary 19 deals with roughly $d^{6}$ input vectors of dimension $d$, while in Lemma 12 we have vectors of dimension $ \approx d^2$: that is why we get a bound of the form $ n \ge d^{12} \dots$
> 2. Please refer to point 3 of the general answer.
> 3. We thank the reviewer for spotting this mistake. We rephrased the sentence in the abstract to specify that we tackle networks of arbitrary depth, and in the main body to credit Malach et al.
> 4. Please refer to the first point of our global reply.
>
> Questions
> 1. We directly approached the CNN case to obtain a more general result. Nonetheless, the structure used in our argument does not produce particularly meaningful results for the MLP case, where one would ideally be able to prove the SLTH for pure neuron pruning. We are currently working on this. Regarding the subset-sum lemma, formulating any type of structured pruning as an instance of the subset-sum problem would require a multidimensional version of the problem. Yet, the specific variant will depend on the target structure.
> 2. Empirical studies with a structured version of edge pop-up [Ramanujan et al., 2020] is a natural and very interesting direction for future work. We also highlight that we provide worst-case bounds and that we do not believe they are optimal. So, much like what we see with unstructured SLTH algorithms, we expect structured versions of them to perform better than the theoretical worst-case guarantees.

---

> > ### Comment · Reviewer_TQCH · 2023-08-12
> > **Thank you for the rebuttal**
> >
> > W1: Sorry for the confusion, but my concern was about the proof sketch in the main paper, not the one in Appendix. I read the proof sketch in the main paper but cannot find any intuition on why the degree of the polynomial order is obtained as $d^{12} c^6 \log(\cdots)$, i.e., the intuition of the degrees $12$ or $6$. Also I cannot find any effort for providing some intuition on the existence proof instead of technical details.
> >
> > W4: Unfortunately, both the paper and the rebuttals finally contain no experimental support for their claim. Since their theoretical result only proves for the case of polynomial over-parameterization, I strength my doubt about the practical validity on the structured version of SLTH. For example, when we want to approximate a CNN with the channel $c=64$ by structured pruning, the proof requires to prune a CNN more than $c^6=68719476736$ times larger.
> >
> > So my understanding on the paper is: while the theoretical result is totally meaningless in practice for now, no experimental supports are provided. Taking these into consideration, unfortunately, I leaned towards reject side because accepting this work may lead the community to a wrong understanding on the structured SLTH.

---

> > > ### Author Response · Authors · 2023-08-13
> > >
> > > We stress that the SLTH was not known to hold for structured pruning and that, since previous techniques lead to exponential bounds, it was possible that it did not. Our work, thus, proves that something that could have been impossible (exponential) is, in fact, possible (polynomial).
> > >
> > > Since the influential contribution of Malach et al. [2020] also provides unfeasible bounds¹, we assume that the reviewer considers our result to be "meaningless in practice" due to the current lack of a structured version of *edge pop-up*. That is, when Malach et al. [2020] was published, methods for finding (unstructured) strong tickets in practice already existed. On the other hand, we offer a theoretical analysis of a problem for which no algorithm was developed yet. However, we do not see why this type of theoretical work would not be worthy. In fact, we are excited by the possibility that our results motivate research on algorithms for finding structured lottery tickets.
> > >
> > > Regarding the possibility that our work could mislead the community, we politely disagree. We claim to prove a version of the SLTH for structured pruning and proceed to present the proof. We do not claim to provide any algorithm for finding strong tickets, let alone an efficient one. We fail to see how this could be misleading. It is also unclear to us how results with a specific type of algorithm would help support our claims. In particular, our theorem applies to essentially all possible parameterizations of the target architecture, including those that cannot be learned by gradient-based methods.
> > >
> > > We sincerely believe that our result, a polynomial bound on a structured version of the SLTH, is of the interest of the theoretical community of NeurIPS.
> > >
> > >
> > > [¹]: For example, applying the bounds from Malach et al. [2020] to a single layer of Lenet-5 (a very small architecture) within error of at most 1% would require increasing the width of that layer by a factor much larger than $\frac{84^4}{0.01^2} = 497871360000$. (To be clear, we are highly appreciative of the work by Malach et al. and we are not trying to diminish its value. We are merely highlighting that the reviewer's argument is not specific to our work.)

---

> > > > ### Comment · Reviewer_TQCH · 2023-08-14
> > > >
> > > > Yes, **Malach et al. [2020] indeed made the first step towards theoretical understanding on the highly feasible phenomenon** that was already observed in Ramanujan et al. [2020], which finally leads to practically meaningful theory in the follow-up works.
> > > > On the other hand, **this submitted work establishes a practically vacuous theory for the highly non-feasible phenomenon of structured SLTH** at least for now, whose unfeasiblity follows from the negative results for the structured SLTH given in Malach et al (Theorem 3.3). Given the lack of practical evidence in this submission, both the title and claims are overclaimed and misleading ones. Therefore, I'm concerned that accepting this work to NeurIPS may lead the community to a misleading conclusion.
> > > >
> > > > [1] Malach et al. "Proving the lottery ticket hypothesis: Pruning is all you need." (2020)
> > > >
> > > > [2] Ramanujan et al. "What's hidden in a randomly weighted neural network?." (2020)

---

> > > > > ### Author Response · Authors · 2023-08-14
> > > > >
> > > > > The result by Malach et al. [2020] focuses on pure neuron pruning of MLPs with 2 layers (one hidden layer). Broadly speaking, the authors show that, under this setup, neuron pruning is equivalent to the random features model. In turn, the latter model is known to be limited as Yehudai and Shamir [2019] proved that approximating even a single ReLU neuron under the random features model can require either exponentially large weights or an exponentially wide hidden layer.
> > > > >
> > > > > However, it is important to note that structured pruning extends beyond pure neuron/filter pruning. Our approach also makes use of block-sparsity, leading to a less constrained setup. With that, diverging from the conclusions of Malach et al. [2020], our theorem asserts that polynomial overparameterization is sufficient to approximate ReLU neurons through structured pruning. Indeed, we show that alternative forms of structured sparsity can be pivotal in overcoming the exponential threshold highlighted in their research.
> > > > >
> > > > > We are certainly open to refining our text to address any passages that might be perceived as exaggerated or misleading. We would appreciate it if the reviewer could highlight those of particular concern.
> > > > >
> > > > > As for the title, the original SLTH states that “within a sufficiently overparameterized neural network with random weights, there exists a subnetwork that achieves competitive accuracy” [Ramanujan et al. 2020]. In the context of our work, given any task $T$, consider a CNN architecture $f$ that is large enough to realise the optimum of $T$ ($f$ serves as reference to make sense of the “overparameterization”). Our results prove that a random CNN $g$ that is sufficiently overparameterized (polynomially, relative to $f$) contain a *structured* subnetwork that approximates $f$. In particular, it achieves competitive accuracy compared with a trained $g$. Thus, we prove that CNNs contain structured strong lottery tickets.
> > > > >
> > > > > Given that, the only way we could think of possibly making our title more precise is to add the polynomial nature of our bounds to it, yielding “Polynomially overparameterized convolutional neural networks contain structured strong lottery tickets”.
> > > > >
> > > > >
> > > > > Yehudai and Shamir [2019]: *On the Power and Limitations of Random Features for Understanding Neural Networks, Gilad Yehudai Ohad Shamir, NeurIPS 2019*.

---

> > > > > > ### Comment · Reviewer_TQCH · 2023-08-14
> > > > > >
> > > > > > > However, it is important to note that structured pruning extends beyond pure neuron/filter pruning. Our approach also makes use of block-sparsity, leading to a less constrained setup. With that, diverging from the conclusions of Malach et al. [2020], our theorem asserts that polynomial overparameterization is sufficient to approximate ReLU neurons through structured pruning. Indeed, we show that alternative forms of structured sparsity can be pivotal in overcoming the exponential threshold highlighted in their research.
> > > > > >
> > > > > > While I agree on this perspective is indeed interesting, I'm still curious about how such a difference between exponential and polynomial overparametrization affects the feasibility of structured SLTH. Nevertheless, the following author's suggestion sounds fair enough:
> > > > > >
> > > > > > > Given that, the only way we could think of possibly making our title more precise is to add the polynomial nature of our bounds to it, yielding “Polynomially overparameterized convolutional neural networks contain structured strong lottery tickets”.
> > > > > >
> > > > > > In addition to fixing the title to something like that (at least it should contain a nuance of polynomial overparameterization), I hope the author to include the above discussion on the relationship between their results and the ones in Malach et al. on structured SLTH. I really appreciate the author's fairness. Although my assessment is still around borderline due to the lack of practical evidence, I would like to raise my score.

---

> > > > > > > ### Author Response · Authors · 2023-08-14
> > > > > > >
> > > > > > > We thank the reviewer for the productive discussion. We are especially grateful to the reviewer for reminding us of the structured pruning portion of the work of Malach et al. In the final version we will adapt the title and discuss the points raised in the exchange with the reviewer.

---

### Official Review · Reviewer_ryBt · 2023-07-03

**Soundness:** 3 good
**Presentation:** 3 good
**Contribution:** 3 good
**Rating:** 7
**Confidence:** 3

**Summary:**

The authors show that structured Strong Lottery tickets are contained within convolutional neural networks.
In order to construct this proof, the authors propose the Multidimensional Random Subset Sum lemma which approximates a target vector by using a set of random vectors which are normally scaled normally distributed.
This results builds on the Subset Sum approximation by Lueker, which is commonly used in proving the Strong Lottery Ticket hypothesis for standard architectures.
The authors further leverage this lemma to approximate a target function with a structurally pruned CNN which is twice as deep and wider by a factor.

Overall this paper provides an important existence proof for strong lottery tickets in structurally pruned CNNs as a first step in trying to understand structured pruning theoretically.

**Strengths:**

1. The authors for the first time have proven the existence of SLTs in structurally pruned networks giving hope that a computationally efficient and trainable SLT might exist that can be leveraged for efficienct on existing hardware (contrary to current empirical results in literature which are worse than unstructured pruning).
2. The authors also extend the result of Lueker for random vectors.
3. The paper is clearly written.

**Weaknesses:**

1. The bounds provided by the authors still require twice the depth in order to approximate a target function. Burkholz et al have proposed a way to use only L + 1 depth to approximate a target function. Is this still possible with structural pruning in CNNs?
2. As mentioned by the authors, the proposed construction looks at structured pruning where entire channels are pruned away. Would block sparsity (eg; N:M sparsity) allow for a better Subset Sum bound potentially i.e. are there specific sparsity patterns which can allow sparser constructions. This might guide us towards a identifying layerwise sparsity ratios for training sparse networks.

**Questions:**

1. The construction requires a layer of width $n c_0$ such that after pruning the width is $m c_0$. These factors would effectively determine the overall sparsity a network can achieve. Would this translate to structured pruning results we observe in practice?



**Limitations:**

The authors provide their construction for ReLU networks. However, Burkholz et al. [1] show existence results for different activation functions.
Do the proposed existence results also transfer to other activations?

[1] Burkholz, Rebekka. "Most activation functions can win the lottery without excessive depth." NeurIPS 2022

---

> ### Author Rebuttal · Authors · 2023-08-09
>
> We are grateful to the reviewer for all the observations pointed out in the review. We try to address the weaknesses, the question and the limitation.
>
> Weaknesses
> 1. We believe it is possible, but so far we could not check all the necessary adaptations. Such improvement requires further investigation.
> 2. In principle, yes. The size of the “structure” to be pruned (be it a filter, a “block” or something else) decides the dimensionality of the Random Subset-Sum Problem to be solved, so, in general, one can expect better bounds from more fine grained pruning. Of course, in the limit, one would obtain the best bounds by considering “blocks” with a single weight, recovering the unstructured pruning case. Thus, one must keep in mind the trade-off we discuss in the paragraph from lines 51-59. Overall, we have not considered $N:M$-sparsity, but we think this is a promising direction for future work.
>
> Questions
> 1. We do not think so. To recapitulate, our results involve transforming a dense convolutional layer with $nc$ filters to obtain a (still dense) one with $mc$ filters, where $m = O(\sqrt{n/\log 1/\epsilon})$, so that the final network is potentially much smaller than the original one (see, e.g., lines 147-150). However, we expect improved bounds on $n$ to bring $m$ closer to $n$.
>
> Limitations
> 1. We do believe the result holds for a more general class of activation functions: however, we do not see a trivial way to prove this and further investigation is required. We remark that, as discussed in the limitations section, we do not think that the technique used by Burkholz [2022] can be immediately adapted to our case because we use the fact that the ReLU function gets rid of negative terms.

---

> > ### Comment · Reviewer_ryBt · 2023-08-12
> > **Acknowldegdement of rebuttal**
> >
> > I thank the authors for the clarifications.

---

### Official Review · Reviewer_LtV9 · 2023-07-09

**Soundness:** 4 excellent
**Presentation:** 4 excellent
**Contribution:** 3 good
**Rating:** 7
**Confidence:** 1

**Summary:**

This paper extends analytical work on the strong lottery ticket hypothesis from unstructured to structured pruning, and shows the applicability to Convolutional Neural Networks.

**Strengths:**

I'm not an expert in this domain, but both the contribution looks strong to me.

Clarity:
The introduction is extensive and appropriately places the current paper into the existing research landscape. I could follow well, even though I'm not directly in this domain. In the analysis section, proofs and other statements are appropriately accompanied by explanatory statements, which makes following them easier.

Quality:
The paper is of high quality. It advances a subfield that is actively being researched.

Originality:
To my knowledge, the work is original. It extends various previous work in a non-trivial manner. The authors place the contributions appropriately among previous results and reference related work.

Significance:
I really cannot accurately estimate the significance of this work. The overall question around whether structured sparsity in CNNs can induce strong lottery tickets is certainly significant. I'm not sure how much the used structures in this work (for example, every other layer being a 1x1 layer) limit the significance of the results.

**Weaknesses:**

- I would have liked to see at least a small experimental section, as the discussed topic lends itself well to empirical evaluation.
- As I said above, just reading the paper gives me very little insight into the meaning and significance of these results, which could be discussed more extensively.

**Questions:**

- In Lemma 8, you deal with variables X_1 through X_{k+j}, but then only condition on Z_1 through Z_k in the text, and Z_1 through Z_n in the statements. It's entirely possible that I misunderstand, but is all of this correct? For example, n is never even introduced as a number.
- I don't quite understand why the n-channel-blocked-masks are needed. In definition 2 you mention that this defines the structured sparsity of the networks, but doesn't the removal of entire filters by itself already constitute structured sparsity?
- Related to above: In Lemma 12 you describe the network including the channel-blocked-masks as the "pruned" network, but from this we still need to remove filters to obtain the smaller network that appropriately approximates K. So which action exactly constitutes the pruning?
- In Lemma 11, you state that V * S-tilde "contains only non-negative edges going from each input channel t [...]". How can that be? V has arbitrary real numbers, and S-tilde is a binary mask. Maybe I've missed something here, I could see that statement be made when including the ReLU in some way, but I'm a bit lost the way it is now.

**Limitations:**

The authors have appropriately addressed limitations in Section 5. Questions on societal impact do not apply here.

---

> ### Author Rebuttal · Authors · 2023-08-09
>
> We thank the reviewer for appreciating our work and the valuable comments. We try to address all concerns.
>
> Weaknesses
> 1. Please refer to point 3 of the general answer.
> 2. We will expand the text in the main body of our document to further discuss the significance of our results by recovering some of the contextualisation from previous works.
>
> Questions
> 1. We apologise for the typo. There, $n = i + k$ and all the shown probabilities are conditional on the outcomes of $X_1, \dots, X_n$. In the supplementary material, the indices in the conditional event are correct (but we forgot to explicitly state $n = i + k$). We updated the main text and the SM accordingly.
> 2. The removal of entire filters does constitute structured sparsity. However, besides choosing which of the (random) filters to keep, we also need to ensure that those are effectively combined in a way that allows us to leverage our MRSSP results to approximate target filters. We achieve this through the $n$-channel-blocked structure. Put loosely, while filter pruning allows us to select subsets (of filters), we need other structures to ensure the subset is “summed” in a suitable way. Of course, the desired operation has many more nuances than a simple sum.
> 3. (and 4) We agree that the text around Lemma 11 is unclear. There are two ways of looking at the pruning in Lemma 11. The mask can be seen as a channel-blocked mask that is applied only to some filters, i.e., we keep filter $k$ if its $t = \lceil \frac{k}{2n} \rceil $-th channel is non-negative and $2t-1 = \lceil \frac{k}{n} \rceil $  or it is non-positive and $2t = \lceil \frac{k}{n} \rceil $. Formally, one can again use a combination of two masks: the first one is a $2n$-channel blocked mask, the second one removes filters. We clarified this explicitly in the statement and proof.
> The pruning in Lemma 12 consists of the actions described in Lemma 11 – the application of the channel blocked mask and the filter removal based on entries’ signs – plus another step of filter removal given by the MRSSP result. We realise that this might cause some confusion, and we added some sentences that explain that all these three parts define the final pruning.  When looking at Lemma 11 and 12 together, one can see a three-stage pruning process: application of $2n$-channel blocked masks (1st stage), filter removal according to entries' signs (2nd stage), filter removal given by the MRSS result (3rd stage). Since masking is both associative and commutative, one can look at the whole process as follows: First, we apply channel blocked masks (1st stage). Second, we remove filters (combination of the 2nd and the 3rd stage).

---

> > ### Comment · Reviewer_LtV9 · 2023-08-22
> >
> > Thank you very much for the considered responses, and for the adjustments.
> > As I said, I have limited knowledge, but in my opinion, the paper should be accepted.

---

### Author Rebuttal · Authors · 2023-08-09

We thank all reviewers for their valuable comments. We are pleased that reviewers appreciated our contribution. Only one of the evaluations leaned towards the negative, and it seemed to be due to misunderstandings that we hope to have solved.

Some reviews brought similar questions, to which we reply below. We also modified the text to further clarify those points.

1. **Our bounds are not optimal**: Directly applying techniques from previous works on the unstructured SLTH to the structured setting would yield exponential bounds (even for the MLP case). Thus, our contribution extends the field qualitatively by showing that polynomial overparameterization suffices. Much like the original polynomial bounds for the unstructured case by Malach et al. [2020] were improved by subsequent works [Orseau et al., 2020; Pensia et al., 2020], we expect our contribution to pave the way for further research and consequent sharper bounds.
2. **We offer a theoretical contribution**: While our work is motivated by the practical concerns underlying structured sparsity, we do not propose a practical algorithm for finding strong lottery tickets. In line with many works related to ours (e.g., Pensia et al. [2020], da Cunha et al. [2022], and Burkholz [2022a,b]), we instead offer theoretical insights about the problem. For example, we establish a baseline for the potential performance of SLTH algorithms. Such a contribution opens the door for valuable future directions, including the extension of SLTH algorithms such as edge pop-up to the structured setting. Finally, our analysis and that of previous works investigate the worst possible case for those algorithms. Thus, they do not set an upper-bound on their practical performance on common architectures.
3. **Experiments are unfeasible for this work**: We agree that experiments would be interesting to see. However, performing meaningful experiments by directly solving subset-sum problem with solvers such as Gurobi (as in, e.g., [da Cunha et al. 2022] or [Pensia et al. 2020]) would be prohibitively expensive as the multidimensional version of the SSP is much harder to solve directly. An alternative would be to extend an algorithm such as edge pop-up [Ramanujan et al., 2020] to our setting (structured pruning). This is a natural and very interesting direction for future work, but it is worth a whole paper on its own.

---

### Decision · Program_Chairs · 2023-09-21

**Decision:**

Accept (poster)

**Comment:**

This paper proposes a version of the "Strong Lottery Ticket Hypothesis" that works for convolutional networks and other forms of structured pruning like block sparsity. The reviewers were generally in favor of acceptance (especially after very productive conversations during the discussion period) and I see no reason to overrule them. I think this is a niche problem with limited applicability, but I respect that it is an interesting problem to many and this paper is a productive contribution to that area.